# Structural insights into the activation of human calcium-sensing receptor

Xiaochen Chen[1†], Lu Wang[1†], Qianqian Cui[1†], Zhanyu Ding[1†], Li Han[1], Yongjun Kou[1], Wenqing Zhang[1], Haonan Wang[1], Xiaomin Jia[1], Mei Dai[1], Zhenzhong Shi[1], Yuying Li[1], Xiyang Li[1], Yong Geng[1,2]*

[1]The CAS Key Laboratory of Receptor Research, Shanghai Institute of Materia Medica, Chinese Academy of Sciences, Shanghai, China; [2]University of Chinese Academy of Sciences, Beijing, China

**Abstract** Human calcium-sensing receptor (CaSR) is a G-protein-coupled receptor that maintains $Ca^{2+}$ homeostasis in serum. Here, we present the cryo-electron microscopy structures of the CaSR in the inactive and agonist+PAM bound states. Complemented with previously reported structures of CaSR, we show that in addition to the full inactive and active states, there are multiple intermediate states during the activation of CaSR. We used a negative allosteric nanobody to stabilize the CaSR in the fully inactive state and found a new binding site for $Ca^{2+}$ ion that acts as a composite agonist with L-amino acid to stabilize the closure of active Venus flytraps. Our data show that agonist binding leads to compaction of the dimer, proximity of the cysteine-rich domains, large-scale transitions of seven-transmembrane domains, and inter- and intrasubunit conformational changes of seven-transmembrane domains to accommodate downstream transducers. Our results reveal the structural basis for activation mechanisms of CaSR and clarify the mode of action of $Ca^{2+}$ ions and L-amino acid leading to the activation of the receptor.

**\*For correspondence:**
gengyong@simm.ac.cn

[†]These authors contributed equally to this work

**Competing interest:** The authors declare that no competing interests exist.

## Introduction

Extracellular calcium ions ($Ca^{2+}$) are required for various kinds of biological processes in the human body. Human calcium-sensing receptor (CaSR) is a G-protein-coupled receptor (GPCR) that senses small fluctuations of extracellular levels of $Ca^{2+}$ ions in the blood (*Brown et al., 1993*). It maintains $Ca^{2+}$ homeostasis by the modulation of parathyroid hormone (PTH) secretion from parathyroid cells and the regulation of $Ca^{2+}$ reabsorption by the kidney (*Brown, 2013*). Recently, it has been reported that CaSR is also a phosphate sensor that can sense moderate changes in extracellular phosphate concentration (*Centeno et al., 2019*; *Chang et al., 2020*; *Geng et al., 2016*). Dysfunctions of CaSR or mutations in its genes may lead to $Ca^{2+}$ homeostatic disorders, such as familial hypocalciuric hypercalcemia, neonatal severe hyperparathyroidism, and autosomal dominant hypocalcemia (*Hendy et al., 2009*; *Pollak et al., 1993*; *Ward et al., 2012*).

CaSR belongs to the family C GPCR that includes gamma-aminobutyric acid B (GABA_B) receptors, metabotropic glutamate receptors (mGluRs), taste receptors, GPRC6a, and several orphan receptors (*Ellaithy et al., 2020*; *Hannan et al., 2018*; *Heaney and Kinney, 2016*; *Pin and Bettler, 2016*). Like most class C GPCRs, CaSR functions as a disulphide-linked homodimer. Each subunit of CaSR is comprised of a large extracellular domain (ECD) that contains a ligand-binding Venus flytrap (VFT) domain and a cysteine rich domain (CRD), and a seven-transmembrane domain (7TMD) that connects to CRD to carry signals from VFT domain to downstream G proteins (*Geng et al., 2016*; *Zhang et al., 2016*).

CaSR can be activated or modulated by $Ca^{2+}$ ions, amino acids (*Geng et al., 2016*; *Liu et al., 2020*; *Zhang et al., 2016*), L-1,2,3,4-tetrahydronorharman-3-carboxylic acid (TNCA), a tryptophan derivative ligand (*Zhang et al., 2016*), and several commercial calcium mimetic drugs, such as cinacalcet (*Leach et al., 2016*; *Nemeth et al., 2004*), etelcalcetide, and evocalcet (positive allosteric modulator, PAM,

of CaSR) that are used for patients with end-stage kidney diseases undergoing dialysis (*Alexander et al., 2015*; *Leach et al., 2016*; *Walter et al., 2013*).

Recent groundbreaking structural studies of several full-length class C receptors, such as mGluR5 (*Koehl et al., 2019*) and GABA_B receptors (*Kim et al., 2020*; *Mao et al., 2020*; *Papasergi-Scott et al., 2020*; *Park et al., 2020*; *Shaye et al., 2020*), by cryo-electron microscopy (cryo-EM) have provided a structural framework to unravel the activation mechanisms of class C GPCRs. The crystal structures of the resting and active conformations of CaSR ECD were solved by two different groups (*Geng et al., 2016*; *Zhang et al., 2016*). More recently, Ling et al. have solved the cryo-EM structures of full-length CaSR in active and inactive states; however, their inactive structures do not show the fully inactive state and exhibit some characteristics of the active conformation of crystal CaSR ECD (*Ling et al., 2021*; *Geng et al., 2016*). In their active structures, they proposed that $Ca^{2+}$ ions and L-Trp work cooperatively to activate CaSR, leading to the closure of VFT domain.

In our study, we used cryo-EM to obtain the structures of full-length CaSR in inactive and agonist+PAM bound conformations. The fully inactive structure is stabilized by a negative allosteric nanobody. In the agonist+PAM bound structure, we identified a new calcium binding site at the inter-domain cleft of VFT, with $Ca^{2+}$ and TNCA constitute a composite agonist to stabilize the closure of the VFT, leading to the conformational changes of the 7TMDs to initiate signaling.

## Results

### Identification of camelid nanobodies stabilizing the inactive state of CaSR

For structural studies, we used nanobody to stabilize CaSR in the inactive conformation. Published structures of CaSR-ECD demonstrate that agonist binding induces conformational changes of VFT

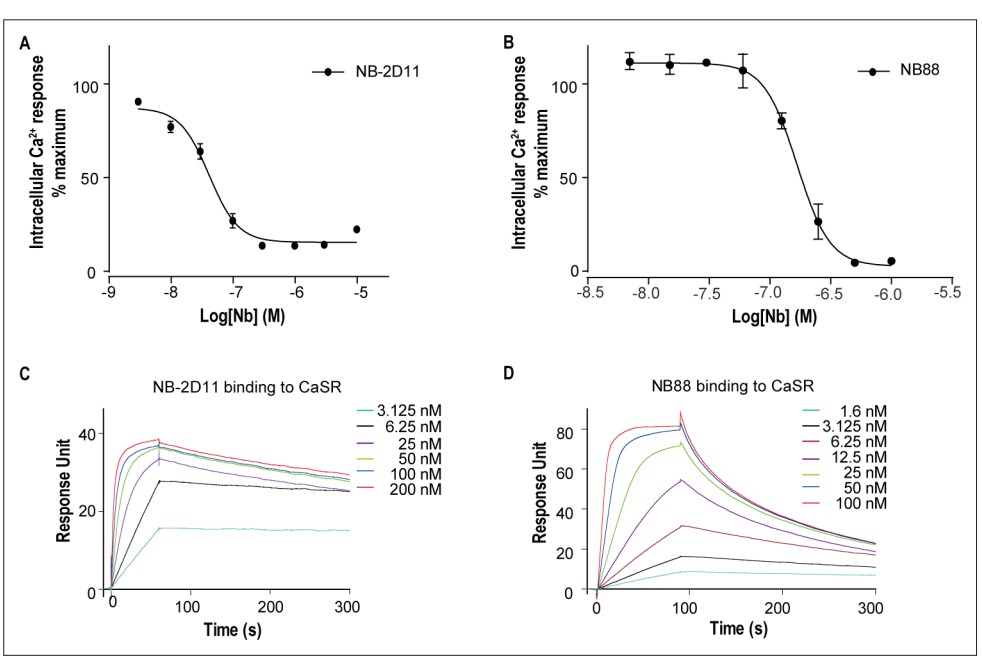

**Figure 1.** The function and binding affinity of NB-2D11 and NB88. (**A**) Dose-dependent NB-2D11-inhibited intracellular $Ca^{2+}$ mobilization in response to $Ca^{2+}$ ions. N = 3, data represent mean ± SEM (*Figure 1—source data 1*). (**B**) Dose-dependent NB-88-inhibited intracellular $Ca^{2+}$ mobilization in response to $Ca^{2+}$ ions. N = 3, data represent mean ± SEM (*Figure 1—source data 1*). (**C**) SPR sensorgram showing that NB-2D11 bound to CaSR with 0.24 nM affinity (*Figure 1—source data 2*). (**D**) SPR sensorgram showing that NB88 bound to CaSR with 3.9 nM affinity (*Figure 1—source data 2*).

The online version of this article includes the following figure supplement(s) for figure 1:

**Source data 1.** Intracellular $Ca^{2+}$ flux assay on CaSR-NB-2D11 and CaSR-NB88 complex.

**Source data 2.** SPR sensorgram of NB-2D11 and NB88 binding affinity.

model of CaSR, whereby two separate LB2 domains approach each other, forming a novel interface in the active state (*Geng et al., 2016*). Based on these structural information, we introduced a potential N-linked glycosylation site on the contacting interface between LB2 domains in the active CaSR to block the interaction of LB2 domains and keep the CaSR in an inactive state. We made a double mutation R227N-E229S at the dimer interface of LB2 domain to introduce N-linked glycosylation at 227 residues site. We immunized two camels with the mutant of CaSR and generated nanobody phage display library. We performed two rounds of bio-panning on the mutant of CaSR and used enzyme-linked immunosorbent assay (ELISA) to verify the nanobodies that specifically bound to CaSR. We performed intracellular $Ca^{2+}$ flux assay to determine whether screened nanobodies could stabilize CaSR in the inactive state. Of the several CaSR binders, NB-2D11 and NB88 significantly inhibited the activity of CaSR with $IC_{50}$ of 41.7 nM and 167.1 nM, respectively (*Figure 1A,B*). Using surface plasmon resonance (SPR) to measure binding kinetics, both nanobodies NB-2D11 and NB88 demonstrated high-affinity binding to CaSR with $K_D$ of 0.24 nM and 3.9 nM, respectively (*Figure 1C,D*). We then selected NB-2D11, which has a greater binding affinity of the two nanobodies, for structural study.

## Determining the cryo-EM structures of full-length CaSR

To obtain the structure of the receptor in the agonist+PAM bound state, we collected a dataset of detergent solubilized full-length CaSR in the presence of PAM cinacalcet, 10 mM calcium and the compound TNCA. We have observed two active conformations with overall resolutions of 2.99 Å and 3.43 Å (*Figure 2—figure supplement 1*). We performed local refinement of ECDs and TMDs separately to obtain maps with resolutions of 3.07 Å and 4.3 Å, respectively, with quality density throughout (*Figure 2—figure supplement 1A*; *Table 1*). The high-quality density maps present well-solved features in the ECD, which allow the unambiguous assignment of calcium, TNCA, and most side chains of amino acids of the receptor (*Figure 2A,C*, *Figure 2—figure supplement 2*). Despite low local resolution of 7TMD, we were able to define the backbone of TM helices and even side chains of some amino acids (*Figure 2C*, *Figure 2—figure supplement 3A*).

To stabilize the structure of CaSR in the inactive state, we collected a dataset of CaSR in glycodiosgenin (GDN) formed micelles in the presence of NPS-2143 (a negative allosteric modulator, NAM) and the inhibitory nanobody (NB-2D11). Cryo-EM data present three conformations of inactive CaSR with an overall resolution of 5.79 Å, 6.88 Å, and 7.11 Å, respectively (*Figure 2—figure supplement 4*). The local refinement focusing on the ECDs and the 7TMDs was performed separately to improve the resolutions to 4.5 Å and 4.8 Å, respectively, with quality density throughout (*Figure 2—figure supplement 4A*; *Table 1*), which enabled us to confidently build the backbone of the inactive CaSR model (*Figure 2D*, *Figure 2—figure supplement 3B*).

The overall structures in the inactive and agonist+PAM bound states are homodimeric arrangement, in which two subunits almost parallelly interact in a side-by-side manner while facing opposite directions. For each subunit, the VFT domain is linked to the canonical 7TMD via CRD, which is almost perpendicular to the bilayer membranes (*Figure 2B,D*). The agonist+PAM bound structure of CaSR displays a substantial compaction compared to the inactive structure, including the reduction of length, height, and width. Moreover, their width changed most obviously because there are four interfaces with interaction between the two protomers at each of LB1 domain, LB2 domain, CRD, and 7TM domains (*Figure 2*), both VFT modules adopt closed–closed conformation, and the TNCA and $Ca^{2+}$ ion composite is bound at the interdomain cleft between LB1 domain and LB2 domain (*Figures 2C and 3A*). The closure of the VFT is relayed to TMD through the interaction of the intersubunit CRD. The overall conformation of our agonist+PAM bound structure is consistent with the recently reported active conformation of the $Ca^{2+}$/L-Trp-bound structure of CaSR (CaSR^Acc) (*Ling et al., 2021*; *Figure 2—figure supplement 5A–D*).

In the inactive structure, there is only one interface at the apex of the receptor and the VFT module adopts an open conformation with the nanobody binding at the left lateral side of each LB2 domain (*Figure 2B,D*). The active state has the overall buried surface area of 3378 Å$^2$, whereas it substantially decreases to 1346 Å$^2$ in the inactive state (*Figure 2—figure supplement 3C,D*). Ling et al. recently published three different structures of CaSR in the inactive state, in which the VFT module adopted closed–closed, open–closed, and open–open conformations. However, due to low resolution, they only built the structure of CaSR in the inactive closed–closed conformation (CaSR^Icc). Comparing our inactive open–open conformation (CaSR^fully inactive) with their CaSR^Icc revealed similar 7TM domains, but

**Table 1.** Cryo-EM data collection, refinement, and validation statistics.

| CaSR | #1 inactive (EMD-30997) (PDB 7E6U) | #2 agonist+PAM (EMD-30996) (PDB 7E6T) |
|---|---|---|
| **Data collection and processing** | | |
| Magnification | 81,000× | 81,000× |
| Voltage (kV) | 300 | 300 |
| Electron exposure (e–/Å$^2$) | 70 | 70 |
| Defocus range (μm) | –1.5 to –2.5 | –1.5 to –2.5 |
| Pixel size (Å) | 1.071 | 1.071 |
| Symmetry imposed | C2 | C2 |
| Initial particle images (no.) | 2,208,402 | 1,546,992 |
| Final particle images (no.) | 1,215,058 | 560,366 |
| Map resolution (Å) FSC threshold | 6.0 0.143 | 3.3 0.143 |
| Map resolution range (Å) | 3.2–7.0 | 2.5–6.5 |
| **Refinement** | | |
| Initial model used (PDB code) | 5k5s, 6n51 | 5k5s, 6n51 |
| Model resolution (Å) FSC threshold | 4.3/5.9/8.0 0/0.143/0.5 | 3.3/3.4/3.7 0/0.143/0.5 |
| Model resolution range (Å) | 4.3–8.0 | 3.3–3.7 |
| Map sharpening $B$ factor (Å$^2$) | –217 | –115 |
| **Model composition** | | |
| Non-hydrogen atoms | 14,214 | 12,751 |
| Protein residues | 1796 | 1592 |
| Ligands | 0 | $PO_4^{3-}$: 2 |
| | | $Ca^{2+}$: 6 |
| | | NAG: 4 |
| | | TNCA: 2 |
| **B factors (Å$^2$)** | | |
| Protein | 102.59/530.90/286.91 | 61.44/302.84/157.67 |
| Ligand | N/A | 91.52/151.96/105.80 |
| **R.m.s. deviations** | | |
| Bond lengths (Å) | 0.002 | 0.002 |
| Bond angles (°) | 0.559 | 0.602 |
| **Validation** | | |
| MolProbity score | 2.5 | 1.49 |
| Clashscore | 14 | 5 |
| Poor rotamers (%) | 0 | 0 |
| **Ramachandran plot** | | |
| Favored (%) | 94 | 97 |
| Allowed (%) | 6 | 3 |
| Disallowed (%) | 0 | 0 |

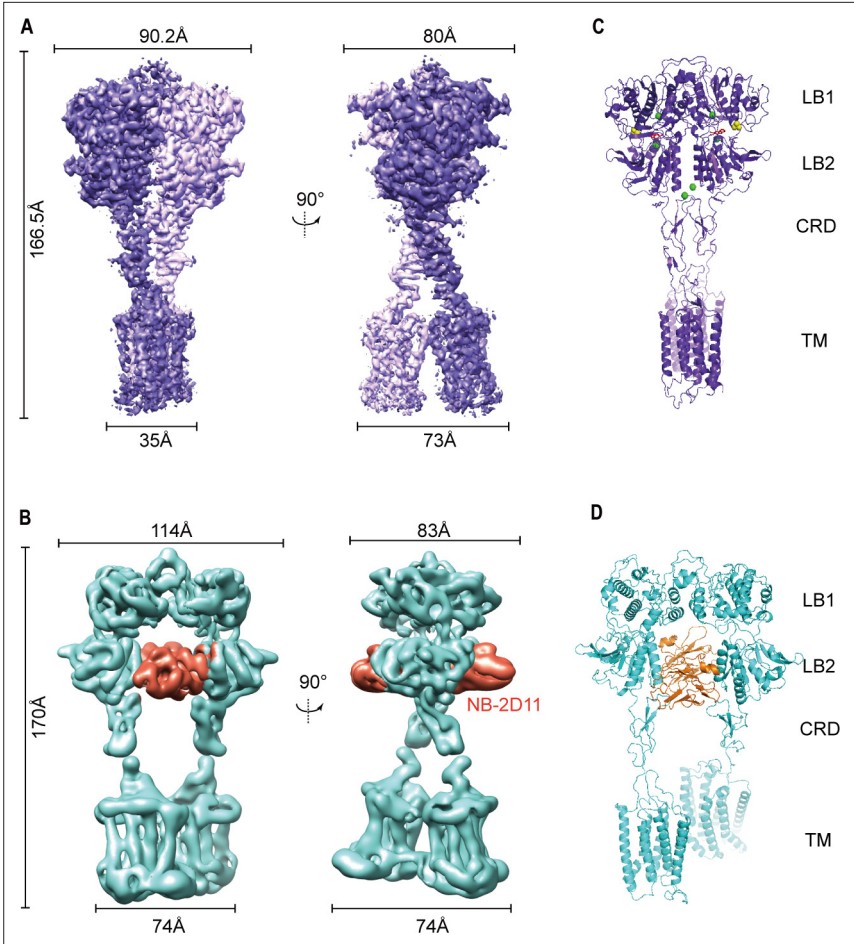

**Figure 2.** Cryo-EM maps and models of full-length CaSR. (**A**) Left panel shows the view of CaSR in the active conformation (purple) from front view, and the right panel shows the view after a 90° rotation as indicated. (**B**) Left panel shows the view of CaSR in the inactive conformation (cyan) bound to NB-2D11 (orange) from front view, and the right panel shows the view after a 90° rotation as indicated. (**C**) Model (Ribbon representation) of CaSR shows the structure of the active state (purple) bound to TNCA (red) and $Ca^{2+}$ ion (green) viewed from the side. (**D**) Model (Ribbon representation) of CaSR shows the structure of the inactive state (cyan) bind with NB-2D11 (orange).

The online version of this article includes the following figure supplement(s) for figure 2:

**Figure supplement 1.** Cryo-EM processing workflow of CaSR bound to agonist+PAM.

**Figure supplement 2.** Agreement between the cryo-EM map of CaSR bound to agonist+PAM and the model.

**Figure supplement 3.** Cryo-EM maps and models of CaSR.

**Figure supplement 4.** Cryo-EM processing workflow of inactive CaSR bound to NB-2D11 in GDN.

**Figure supplement 5.** Comparisons of the structures of CaSR in different conformations.

two totally different VFT module conformations, with their closed–closed conformation presenting similar characteristics to the active state (*Figure 2—figure supplement 5*). This indicates that the CaSR in the inactive state has conformational heterogeneity. In other words, this suggests that in addition to the full inactive state and the active state, there are multiple intermediate states in the process of activation.

## $Ca^{2+}$ and TNCA as a composite agonist activate the full-length CaSR dimer

The cryo-EM map of active state presents a distinct density at the ligand-binding cleft of each protomer, which enabled us to unambiguously model TNCA (*Figure 3A,B*). The binding details of TNCA were the same as previously reported data (*Zhang et al., 2016*). The interactions between

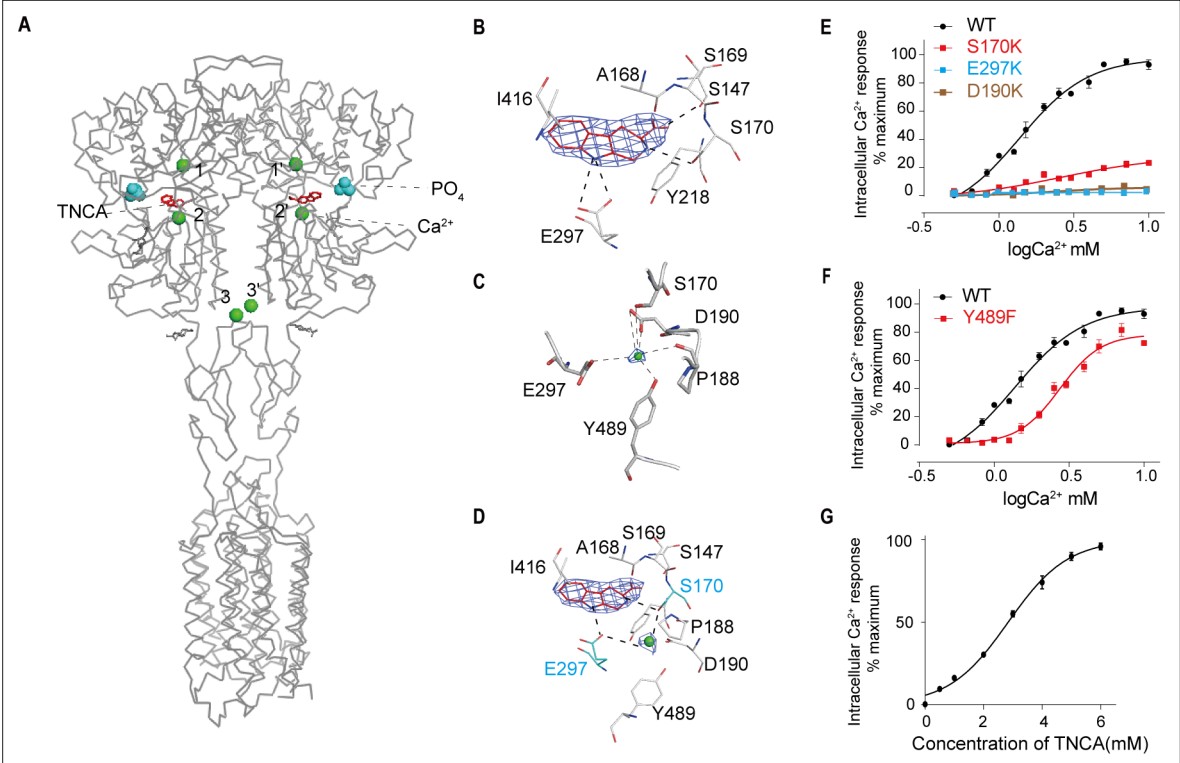

**Figure 3.** Ca$^{2+}$ and TNCA as a composite agonist activate the full-length CaSR dimer directly. (**A**) Ribbon representation of the active CaSR (gray), showing the location of the Ca$^{2+}$-binding sites (green sphere) and TNCA (red). (**B**) Specific contacts between CaSR (gray) and TNCA (red space-filling model), mesh represents the final density map contoured at 17σ surrounding. (**C**) Specific interactions between CaSR and newly identified Ca$^{2+}$ ion (green sphere), the mesh represents the cryo-EM density map contoured at 6.0σ surrounding Ca$^{2+}$. (**D**) Highlighting the newly identified Ca$^{2+}$ and TNCA sharing two common binding residues S170 and E297 (cyan space-filling model). (**E**) Dose-dependent intracellular Ca$^{2+}$ mobilization expressing WT (black dots), mutant S170K (red dots), E297K (cyan dots), and D190K (brown dots) of CaSR. The single mutations were designed based on Ca$^{2+}$ binding sites. N = 4, data represent mean ± SEM (**Figure 3—source data 1**). (**F**) Dose-dependent intracellular Ca$^{2+}$ mobilization expressing WT (black dots), mutant Y489K (red dots) of CaSR. The single mutation was designed based on Ca$^{2+}$ binding sites. N = 4, data represent mean ± SEM (**Figure 3—source data 1**). (**G**) Dose-dependent TNCA-activated intracellular Ca$^{2+}$ mobilization in response to 0.5 mM Ca$^{2+}$ ions. N = 3, data represent mean ± SEM (**Figure 3—source data 2**).

The online version of this article includes the following figure supplement(s) for figure 3:

**Source data 1.** Intracellular Ca$^{2+}$ flux assay on CaSR mutations.

**Source data 2.** Intracellular Ca$^{2+}$ flux assay on CaSR-TNCA complex.

**Figure supplement 1.** Cell surface expression.

TNCA and VFT are primarily mediated by hydrogen bonds (**Figure 3B**). The high-resolution density of active state map enabled us to identify three distinct Ca$^{2+}$-binding sites within ECD of each protomer (**Figure 3A**). Two sites were previously reported (**Geng et al., 2016**; **Ling et al., 2021**), while a new Ca$^{2+}$-binding site was found at the interdomain cleft of the VFT module that is close to the hinge loop and abuts the TNCA binding site, and interacts with both LB1 and LB2 domains to facilitate ECD closure (**Figure 3A–D**). The bound Ca$^{2+}$ ion is primarily coordinated with side chains of D190 and E297, carbonyl oxygen atoms of P188 backbone, and hydroxyl groups of S170 and Y489. Residues P188, D190, and S170 are located in LB1 domain, while E297 and Y489 are in LB2 (**Figure 3C,D**). The main coordination residues (S170, D190, and E297) of the Ca$^{2+}$ ion are consistent with those previously reported (**Liu et al., 2020**). The maps obtained by cryo-EM imaging are insufficient to confirm that the observed density corresponds to calcium. We assume that the density represents the presence of Ca$^{2+}$ based on the following reasons. First, from its hexavalent coordination (coordinating residues P188, D190, S170 and E297, and Y489), this metal is most likely to be Ca$^{2+}$, although another ion cannot be ruled out. Second, we prepared the CaSR sample in a purification buffer supplemented with 10 mM Ca$^{2+}$ and without any other bivalent cation prior to cryo-EM imaging. Third, the main binding

residues (S170, D190, and E297) of Ca²⁺ ion were previously reported (*Liu et al., 2020*), and that single mutation of these residues (D190K, S170K and E297K, and Y489F) significantly reduced the effect of Ca²⁺-stimulated intracellular Ca²⁺ mobilization in cells (*Figure 3E*). The cell surface expression levels of these mutants are all above 80 % compared to the wild-type level (*Figure 3—figure supplement 1*). Finally, mutant of a residue that bind L-amino acid (S147A) also largely impaired the Ca²⁺ effect (*Geng et al., 2016*), indicating the presence of L-amino acid near Ca²⁺ ion and that Ca²⁺ activates CaSR through the L-amino acid.

The Ca²⁺ ion interaction with both the LB1 and LB2 domains implies that it also contributes to the closure of the VFT module. The mutation of residue Y489 on LB2 that is in contact with Ca²⁺, but not L-amino acid, significantly reduces the effect of Ca²⁺-stimulated intracellular Ca²⁺ mobilization in cells (*Figure 3F*). This indicates that Ca²⁺ on its own is very important for stabilizing the closure of VFT, consistent with findings by *Liu et al., 2020*. Ling et al. tried to determine the cryo-EM structures of CaSR in the presence of a high concentration of Ca²⁺ to address the question of whether Ca²⁺ ions alone can activate CaSR in the absence of L-Trp. However, they did not obtain the closed conformation of VFT that only contain the Ca²⁺ ion between the cleft. This result indicates that Ca²⁺ ion alone is insufficient to induce the closure of the VFT module even in the presence of a high concentration of Ca²⁺ ions (*Ling et al., 2021*).

Our structure shows that TNCA bind at the interdomain of VFT module (*Figure 3D*, *Figure 2—figure supplement 2B*), corresponding to the L-amino acid binding site in other class C GPCRs, such as mGluRs and GABA_B receptors. However, it has been reported that Ca²⁺ ion can activate the receptor on its own in various functional assays (*Jensen and Brauner-Osborne, 2007*; *Liu et al., 2020*; *Quinn et al., 2004*; *Saidak et al., 2009*) and L-amino acids enhance the sensitivity of CaSR to Ca²⁺ ion (*Conigrave et al., 2000*; *Liu et al., 2020*). While L-amino acids and their analogies are generally considered PAMs but not agonists of CaSR, they are the endogenous agonists of other class C GPCRs. This is somewhat inconsistent from the perspective of GPCR classification and evolution.

Our CaSR^agonist+PAM structure reveals that the interaction of TNCA with the LB1 and LB2 domains can promote the closure of VFT module, which is a crucial step of the activation for Class C GPCR. The single mutation of the TNCA or L-Trp binding residues (T145I, S147A, S170A, Y218S, E297K) largely impaired the function of the receptor (*Figure 3E*; *Geng et al., 2016*). This suggests that the TNCA or L-Trp plays an important role during the activation of CaSR. Using intracellular Ca²⁺ flux assays, we

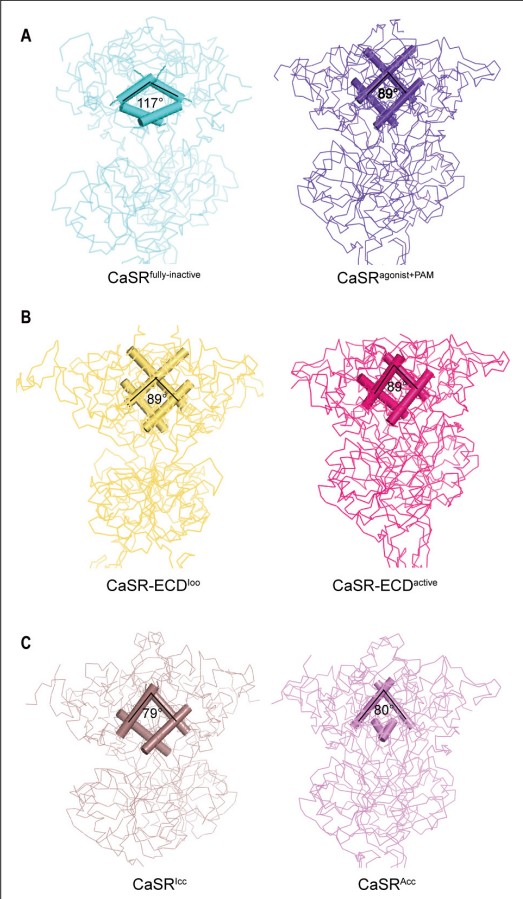

**Figure 4.** Comparisons of intersubunit LB1 domains interfaces in the inactive and active states of CaSR. (**A**) Left panel: The Cα trace of VFT module of CaSR^fully inactive cryo-EM structure (cyan). The B-C Helix angle is 117°. Right panel: The Cα trace of VFT module of CaSR^agonist+PAM cryo-EM structure (purple). The B-Helix angle is 89°. (**B**) Left panel: The Cα trace of VFT module of crystal structure of CaSR-ECD^Ioo (yellow) (PDB:5K5T). The B-Helix angle is 89°. Right panel: The Cα trace of VFT module of CaSR-ECD^active crystal structure (red) (PDB:5K5S). The B-Helix angle is 89°. (**C**) Left panel: The Cα trace of VFT module of CaSR^Icc cryo-EM structure (brown) (PDB:7DTW). The B-Helix angle is 79°. Right panel: The Cα trace of VFT module of CaSR^Acc cryo-EM structure (lavender) (PDB:7DTV). The B-Helix angle is 80°.

The online version of this article includes the following figure supplement(s) for figure 4:

**Figure supplement 1.** Comparisons of intersubunit LB1 domains interfaces in the inactive and active states of CaSR.

found that TNCA directly activated CaSR in the presence of 0.5 mM of $Ca^{2+}$ ions and that the effect on CaSR was concentration-dependent with $EC_{50}$ of 2.839 mM (*Figure 3G*), in agreement with previous reports that L-Trp directly stimulated intracellular $Ca^{2+}$ mobilization in cells stably expressing CaSR using single-cell intracellular $Ca^{2+}$ microfluorimetry (*Geng et al., 2016*; *Rey et al., 2005*; *Young and Rozengurt, 2002*).

It is interesting that our structure shows that the bound $Ca^{2+}$ and TNCA share three common binding residues S170, D190, and E297 (*Figure 3D*). Our experiment has shown that each of single mutations S170A, D190K, and E297K abolishes $Ca^{2+}$-dependent receptor response (*Figure 3E*), consistent with *Liu et al., 2020*. Previous studies have suggested that the extracellular $Ca^{2+}$ increases L-Trp binding (*Geng et al., 2016*), and L-Trp also directly stimulates intracellular $Ca^{2+}$ mobilization through CaSR (*Conigrave et al., 2004*; *Rey et al., 2005*; *Young and Rozengurt, 2002*) and the efficacy and potency of L-Trp increase with increase in $Ca^{2+}$ concentration (*Geng et al., 2016*). As mentioned above, L-amino acids increase the effect of $Ca^{2+}$ ions on the CaSR (*Jensen and Brauner-Osborne, 2007*; *Liu et al., 2020*; *Quinn et al., 2004*; *Saidak et al., 2009*), and TNCA potentiate the $Ca^{2+}$ activity (*Zhang et al., 2016*). Altogether, we show that CaSR is synergistically activated by the composite agonist composed of TNCA and $Ca^{2+}$ ions.

## The conformational transition of the LB1 prepares for the ligand binding during the activation of CaSR

Both inactive and active structures reveal that the interface of LB1–LB1 dimer is predominantly a hydrophobic core, which is formed by the residues on two central helices (B and C) of each protomer, including V115, V149, as well as L156 for inactive structure and L112, L156, L159, and F160 for active crystal structure (*Figure 4A*, *Figure 4—figure supplement 1A,B*). On the dimer interfaces, The B–C helix angle has rotated approximately 28° from inactive state (117°) to active state (89°) (*Figure 4A*).

Our result is in line with earlier reports of CaSR and other class C GPCRs activation. Liu et al. detected reorientation of LB1–LB1 dimer during activation using a FRET-based conformation CaSR sensor (*Liu et al., 2020*). mGluR5 receptor changes from active to apo state with an approximately 59° rotation of the B–C helix angle (*Koehl et al., 2019*; *Figure 4—figure supplement 1C,D*).

We then compared the B-C helix angles of previously reported CaSR structures, including CaSR-ECD[loo], CaSR-ECD[active], CaSR[Icc], and CaSR[Acc]. No rotation of B–C helix was observed between inactive CaSR-ECD[loo] and active CaSR-ECD[active] crystal structures (*Figure 4B*; *Geng et al., 2016*), despite changing VFT conformation from closed–closed to open–open. Similarly, there was only a small rotation of 1° between CaSR[Icc] and CaSR[Acc] (*Figure 4C*; *Ling et al., 2021*). VFT module of the inactive CaSR[Icc] adopts a closure conformation, and L-amino acid binds at the interdomain of the VFT (*Ling et al., 2021*); both features are characteristics of an active state. The B–C Helix angle of all four reported structures (CaSR-ECD[loo], CaSR-ECD[active], CaSR[Icc], CaSR[Acc]) resemble that of our active CaSR[agonist+PAM] structure (*Figure 4A–C*).

The B–C helix of the recently reported CaSR structure in the inactive states is same as that of our CaSR[agonist+PAM], although the 7TMD is similar to that of the inactive state (*Ling et al., 2021*; *Figure 4C*). The VFT module of the reported CaSR[Icc] adopts the closure conformation; moreover, the L-amino acid binds at the interdomain of the VFT (*Ling et al., 2021*), which are features of the active state. The rotation of B–C helix is not observed in both the active crystal structures of CaSR ECDs with the closed–closed conformation of VFT module (PDB: 5K5S) and the inactive crystal structure with the open–open conformation of VFT module (PDB: 5K5T), In summary, we propose that these reported conformations should be considered intermediate states in the activation process of CaSR because they exhibit some characteristics of the active state. In our inactive cryo-EM structure, the B–C helix angle is similar to that of mGluRs in the inactive state, with the VFT domain adopting an open–open conformation (*Figure 4A*, *Figure 4—figure supplement 1C,D*). Therefore, our inactive cryo-EM structure represents the full inactive state. We hereby designate our inactive structure as CaSR[fully inactive].

The LB1 domain plays a predominant role for anchoring ligands. Superimposition of LB1 domains of inactive (CaSR[fully inactive]), intermediate (CaSR-ECD[loo]), and agonist+PAM bound (CaSR[agonist+PAM]) conformations, reveals that our inactive conformation has a significantly different LB1 structure compared to the intermediate conformation (*Figure 4—figure supplement 1E*), whereas the LB1 domains in the intermediate and agonist+PAM bound states are well superimposed with a backbone r.m.s.d. of

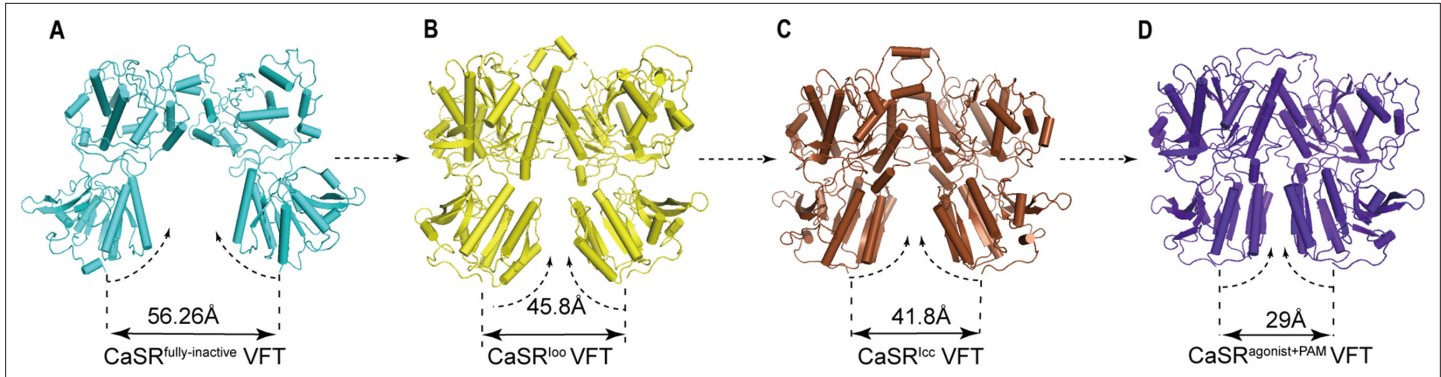

**Figure 5.** The conformational changes of LB2 domains in three states. (**A**) The CaSR^fully inactive (cyan) conformation of VFT module. The distance between C termini of the two LB2 domains is 56.26 Å. (**B**) CaSR-ECD^loo (yellow) (PDB:5K5T) conformation of VFT module. The distance between C termini of the two LB2 domains is 45.8 Å. (**C**) CaSR^lcc (brown) conformation of VFT module (PDB:7DTW). The distance between C termini of the two LB2 domains is 41.8 Å. (**D**) CaSR^agonist+PAM (purple) conformation of VFT module. The distance between C termini of the two LB2 domains is 29 Å.

The online version of this article includes the following figure supplement(s) for figure 5:

**Figure supplement 1.** Superposition of LB1 and VFT domains of CaSR.

0.806 Å (*Figure 4—figure supplement 1F*). Thus, the conformational transition of the LB1 domain from inactive to intermediate state provides the structural basis for ligand binding.

## Spontaneous proximity of LB2 domains during the activation

No significant difference of the overall LB2 conformations is observed among the superposition of inactive, intermediate (CaSR-ECD^loo), and agonist+PAM bound structures (*Figure 5—figure supplement 1A,B*). The cryo-EM structure of CaSR in inactive state displays a relatively large backbone separation distance of 56.26 Å between the C-terminal ends of N541 of each LB2 domain, while it reduces to 45.8 Å in the CaSR^loo state and 41.8 Å in the CaSR^lcc state. A further reduction to 29 Å is observed upon activation in the active model (*Figure 5*). Thus, the two LB2 domains gradually approach each other until they interact, a process that is not induced by the agonists (*Figure 5*, *Figure 5—figure supplement 1C–E*).

## NB-2D11 blocks the interaction of LB2 domains to lock the CaSR in the full inactive conformation

The inactive structure reveals that NB-2D11 binds the left lateral of each LB2 domain from orthogonal view (*Figure 6*), with the hydrophilic interaction interface between the amino acids D53, D99, W102, R101, and E110 from CDR1 and CDR3 of the nanobody and the residues R220, S240, S244, Y246, S247, and E251 from Helix F and Strand I (*Figure 6C*). Superposition of the inactive and agonist+PAM bound LB2 domains shows that NB-D211 occupies the spatial position of the LB2 domain of the other protomer, which blocks the approach of another corresponding subunit LB2 (*Figure 6D*). Our results indicate that the interactions of both LB2 domains are required to activate CaSR, which is the explanation of the inhibitory function of NB-2D11.

## The rotation of LB2 domain propagates to large-scale transitions of the 7TMDs from TM5-TM6-plane to TM6-driven interface

The closure of VFT displays an inward rotation of each LB2 followed by moving upward individually (*Figure 5*). Afterwards, two intersubunit interfaces are formed at the downstream of subunits, including the interaction between the LB2 linked CRDs, which is consistent with the reported crystal structure of CaSR ECD (*Geng et al., 2016*; *Figure 7—figure supplement 1*), and the intersubunit interaction between TMDs (*Figure 7A–F*).

The alignment of individual 7TMD of both inactive and agonist+PAM bound states presents that the helices are well superposed (*Figure 7D*). Although NAM and PAM were added during the preparation of inactive and active samples, respectively, no density of them was observed on the maps due to low resolutions. The inactive structure reveals that TM5 and TM6 constitute a 7TMD plane–plane interface

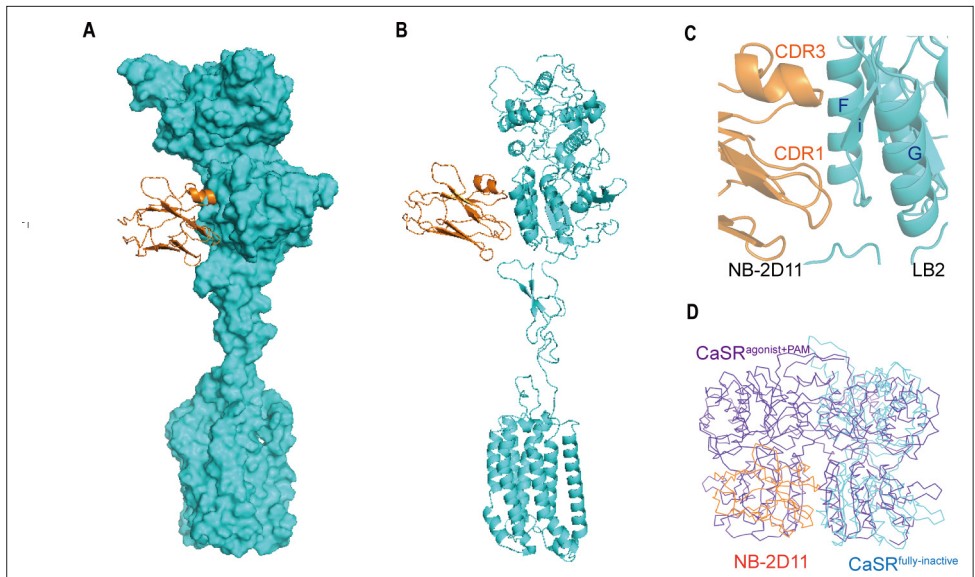

**Figure 6.** The NB-2D11 blocks the interaction of LB2 domains. (**A**) Structure of the inactive CaSR protomer (surface representation, cyan) with NB-2D11 (ribbon diagram, orange) from front view. (**B**) The NB-2D11 (orange) binds the left lateral of the LB2 (cyan) from the front view of the protomer. (**C**) NB-2D11 binds the LB2 domain through a series of polar interactions through CDR1 and CDR3 of the nanobody and the Helix F and Strand I of the CaSR. (**D**) Superposition of NB-2D11 (orange) binding inactive conformations (cyan) and active (purple) conformations based on the LB2 domain of VFT module, showing the whole NB-2D11 in the inactive state crashes with the LB2 domain of another VFT module in the active state.

(*Figure 7E*). There are pairwise symmetrical undefined maps that link the extracellular and intracellular part of TM5 and TM6 in the 7TMD interface (*Figure 7—figure supplement 2A*). Our structure shows a TM5–TM6/TM5–TM6 interaction that it is slightly different from the TM4–TM5/TM4–TM5 plane–plane interaction in the 7TMD interface proposed by *Liu et al., 2020*. Our experiments show that each of the single mutations F789A or F792A attenuates $Ca^{2+}$-induced receptor activity, indicating that this contact plays a role in the activation of CaSR (*Figure 7H,I*). The cell surface expression levels of these mutants are all over 100 % compared to the wild-type level (*Figure 3—figure supplement 1*).

The agonist+PAM bound structure shows a TM6–TM6 interface, contacting at the apex of TM6 helices, which is a hallmark of GPCR activation (*Koehl et al., 2019*; *Figure 7B,F*). To further validate the role of this interface, mutation to P823 in TM6 markedly reduced $Ca^{2+}$-induced receptor activity (*Figure 7J*), indicating that the TM6–TM6 interface is crucial to CaSR activation, consistent with previous studies. Liu et al. reported an interface mediated by TM6 in their active CaSR structure and showed that a cysteine cross-linking at residue A824[6.56] in TM6 led to a constitutively active receptor (*Liu et al., 2020*). Similarly, active mGluR5 (*Koehl et al., 2019*) and GABAB receptors (*Kim et al., 2020*; *Mao et al., 2020*; *Papasergi-Scott et al., 2020*; *Shaye et al., 2020*) have the same TM6–TM6 interface (*Figure 7F*; *Figure 7—figure supplement 2B,C*). TM6 cross-linked mGluR5 and TM6-locked mGluR2 were activated continuously (*Koehl et al., 2019*; *Xue et al., 2015*).

Superposition of inactive and agonist+PAM structures shows a high degree of structural overlap in 7TM domains, with the exception of a bundle comprising of extracellular loop 2 (ECL2) and a stalk linking CRD and TM1 showing slight structural dissimilarity. CRD appears semi-rigid (*Figure 7C*). Therefore, a small rotation of LB2 domains could propagate to large-scale transitions of the TMDs through the CRDs, thereby reorientating the 7TMDs from the inactive plane–plane interface mediated by TM5 and TM6 to the active interface driven by TM6 (*Figure 7E–G*). The proximity of 7TMDs is observed during the activation, from a plane–plane distance of 24 Å in inactive state to 5.7 Å at P823[6.55] in the active state (*Figure 2—figure supplement 3A,B*).

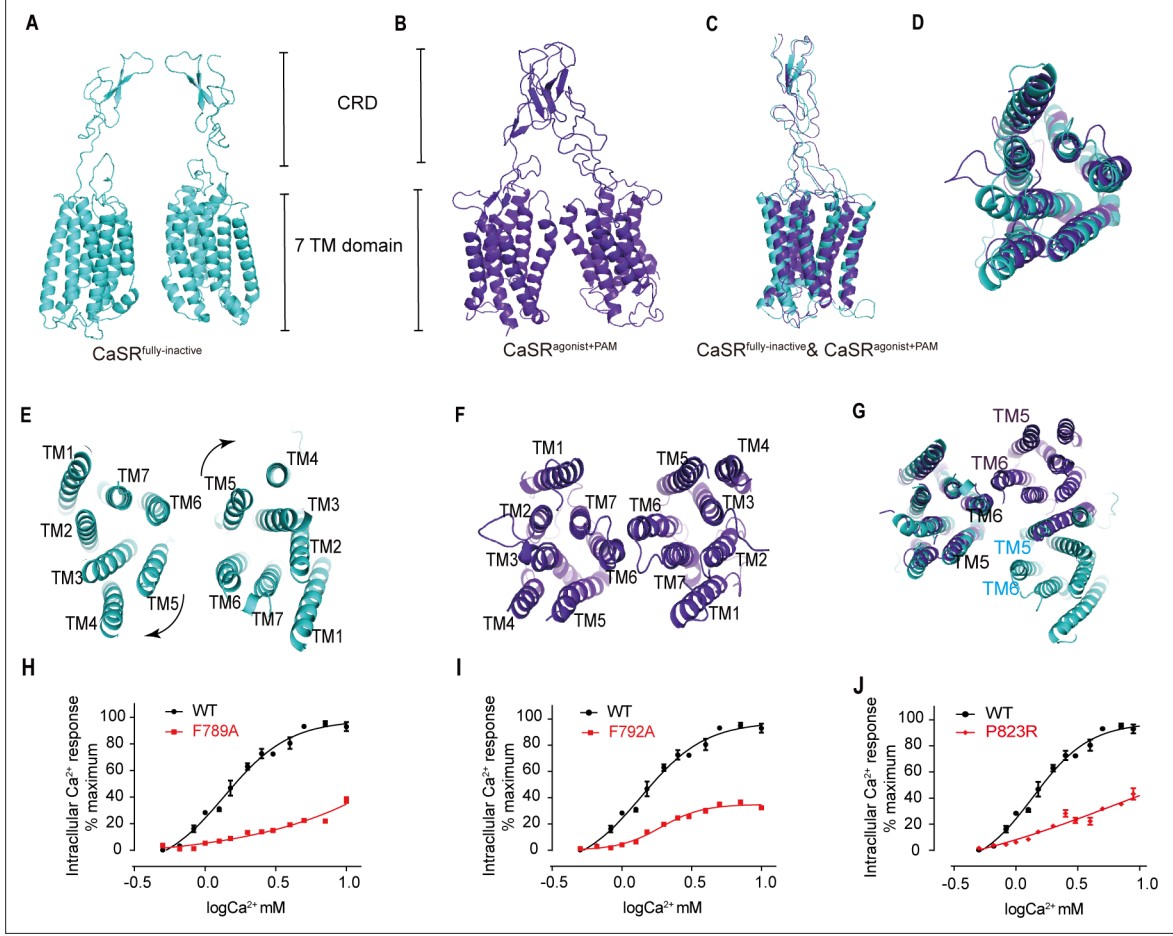

**Figure 7.** The closure of VFT leading to the rearrangement of inter-7TMDs. (**A**) Front view of CaSR^fully inactive CRDs and 7TMDs (cyan). (**B**) Front view of CaSR^agonist+PAM CRDs and 7TMDs (purple). (**C**) The alignment of the part of CRD and 7TMDs in both fully inactive and agonist+PAM bound CaSR. (**D**) The alignment of inactive and agonist+PAM bound 7TMDs from top view. (**E–G**) The 7TMDs interface in the fully inactive state of CaSR is mediated by TM5 and TM6 (cyan) from top view and that of the agonist+PAM state is driven by TM6 from top view. Superposition of 7TMD of the inactive (cyan) and agonist+PAM bound CaSR (purple) show the rotation of 7TMDs. (**H–J**) Dose-dependent intracellular $Ca^{2+}$ mobilization expressing WT (black dots) and mutant (red dots) CaSR (*Figure 7—source data 1*). The single mutations of F789A (**H**), F792A (**I**), and P823R (**J**) were designed based on the inactive density map. For (**H–J**), N = 4, data represent mean ± SEM.

The online version of this article includes the following figure supplement(s) for figure 7:

**Source data 1.** Intracellular $Ca^{2+}$ flux assay on various CaSR mutations.

**Figure supplement 1.** The homodimer interface of the CRDs in the active state of CaSR.

**Figure supplement 2.** Conformational change of the 7TMDs interface during activation.

## Upward movement of LB2 converted into the intra-7TM rearrangement through ECL2

Models of both inactive and active structures reveal that there is a bundle of structure in the junction region between extracellular and transmembrane domain, which is composed of C-terminal elongated peptide of CRD and the twisted hairpin loop of ECL2 (*Figure 8A,B*). Unlike mGluR5 and GABA_B receptors (*Kim et al., 2020*; *Koehl et al., 2019*; *Mao et al., 2020*; *Papasergi-Scott et al., 2020*; *Park et al., 2020*; *Shaye et al., 2020*), which are formed by a twisted three-strand β-sheet, the junction of CaSR is more flexible than that of mGluR5 and GABA_B receptors. The structure of the agonist+PAM bound state shows that the residues 759–763 fragment of ECL2 and the C-terminal residues of the CRD (601–604) form a new interface (*Figure 8A*), which presents a more compactible interaction in the agonist+PAM bound state (*Figure 8B*). In addition, there is another interface involving the residues E759 at the apical loop of ECL2 and the residues W590 at the bottom of the loop composed of

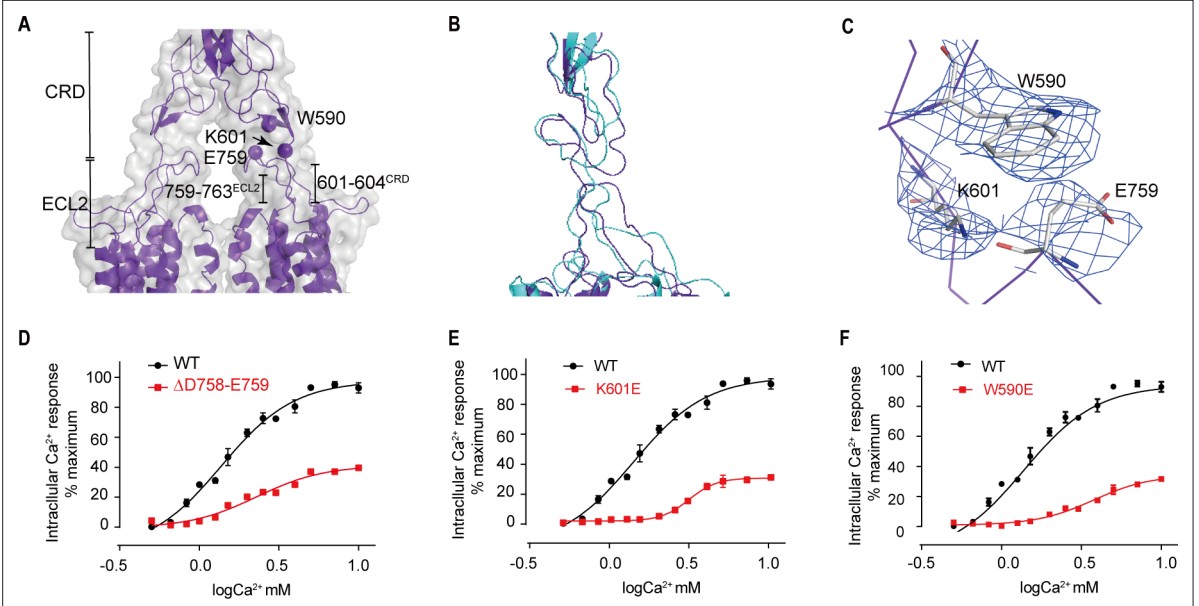

**Figure 8.** Upward movement of LB2 is converted into the intra-7TM conformational rearrangement through ECL2. (**A**) Model in CaSR^agonist+PAM state (purple) and cryo-EM map (gray) showing the contact between the CRD and the ECL2 of the 7TMDs. Critical residues at this interface are shown as spheres at their Cα positions. (**B**) Superposition of the interface between the CRD and the ECL2 of the 7TMD between both fully inactive and agonist+PAM bound conformations. (**C**) Specific contacts between the loop of CR domain and the loop of ECL2 to shift the ECL2 up. (**D–F**) Deletion of residues D758 and E759 (**D**), the single mutation of K601E (**E**) and W590E (**F**) significantly reduced Ca²⁺-induced receptor activity. (WT in black dots and mutant in red dots). For (**D–F**), N = 4, data represent mean ± SEM (**Figure 8—source data 1**).

The online version of this article includes the following figure supplement(s) for figure 8:

**Source data 1.** Intracellular Ca²⁺ flux assay on CaSR mutations.

residues 589–591 for agonist+PAM bound state. In the agonist+PAM bound state, the loop of ECL2 is pulled up by the interaction among E759, W590, and K601, leading to the movement of ECL2 (**Figure 8**), which would raise the reorientation of TM5 and TM6 domains during the activation of CaSR (**Figure 7E,F**). To confirm the importance of this interaction, deletion of residues D758 and E759 at the apex of ECL2 (**Figure 8D**), as well as single mutations of K601E and W590E (**Figure 8E,F**), disrupted these contacts and led to a significantly reduced Ca²⁺-induced receptor activity, Therefore, ECL2 plays a key role in relaying the conformational changes of VFT to the intrasubunit TM domain to rearrange the structure to adapt to downstream transducers, such as G proteins. The cell surface expression levels of Δ758–759 mutant was comparable to that of WT, while W590E and K601E mutants were expressed on the cell surface at approximately 40–50% of WT level (**Figure 3—figure supplement 1**).

## Discussion

In this study, we have determined the cryo-EM structures of CaSR in the fully inactive and agonist+PAM bound states. During the preparation of our manuscript, several CaSR structures have been reported, including structures of closed–closed conformation in the inactive state (CaSR^Icc and CaSR^Trp) and closed–closed conformation of Ca²⁺/Trp bound state (CaSR^Acc and CaSR^Ca). Open-closed and open–open conformations (CaSR^Ioc and CaSR^Ioo) have also been observed; however, they were not built due to low resolution (**Ling et al., 2021**). The overall conformation of our CaSR^agonist+PAM structure is almost identical to that of CaSR^Acc, while the open–open conformational VFT module in our inactive structure (CaSR^fully inactive) is different from the closed–closed conformation in their reported inactive CaSR^Icc structures. In addition, the main conformation changes of CaSR during activation were also described (**Liu et al., 2020**). Complemented with solved crystal structures of CaSR ECD and full-length cryo-EM structures of other class C GPCR, these recent findings allow us to understand the structural framework and essential events that occur during the activation of CaSR. The overall structures of CaSR resemble the recently published structures of mGluR5

and GABA$_B$ receptors, indicating that the structural mechanism of class C GPCRs is similar (*Kim et al., 2020*; *Koehl et al., 2019*; *Mao et al., 2020*; *Papasergi-Scott et al., 2020*; *Park et al., 2020*; *Shaye et al., 2020*).

Multiple structural and functional studies of class C GPCR have demonstrated that there are two typical conformational changes in the VFT domains during receptor activation. One is that the B–C helix angle at the interfaces of LB1–LB1 dimer sharply rotated from inactive to active state. This conformational transition of the LB1 domain is conducive to ligand binding, which is a prerequisite for receptor activation (*Figure 5—figure supplement 1A,B*). For example, increased glutamate affinity and occupancy in mGluR2 active conformation were observed by the mutations in B-Helix (*Levitz et al., 2016*). Another is that the conformation of VFT domain is converted from open to closed for the change of interdomain in one protomer, which is a landmark event during receptor activation.

We used an inhibitory nanobody to stabilize the conformation of CaSR in the inactive state. Our inactive structure shows that the B-C helix angle is about 117° and VFT domain adopts an open–open conformation (*Figure 4A*). The B–C helix angle has rotated approximately 28° from inactive to agonist+PAM bound state and the VFT domain rearranged from open–open configuration to closed–closed configuration during CaSR activation (*Figures 4A and 5*), consistent with other class C GPCR activation mechanism findings. In addition, Liu et al. developed a FRET-based conformation sensor for CaSR through fusion of SNAP-tag at its N-terminus of CaSR subunit to label with fluorophores. Their data showed that the CaSR dimer underwent a large conformational change of LB1–LB1 dimer during activation, in which the B–C helix angle rotated from inactive to active state as the fluorophores labeled the N-terminus of LB1 domain (*Liu et al., 2020*). We used nanobody NB-2D11 to block the proximity of LB2 domains, thus locking the CaSR in an inactivate state (*Figure 6*). Altogether, our structural and functional assay data suggest that our inactive cryo-EM structure represents the full inactive state of CaSR.

The rotation of LB1–LB1 domains is a watershed between inactive and intermediate states. We have reported that the crystal structure of CaSR-ECD in the open-open state (CaSR$^{loo}$) has the same B–C helix angle as that of the active state (CaSR$^{agonist+PAM}$), with LB1 domain configuration that is ready for L-amino acid and Ca$^{2+}$ binding, in contrast to our cryo-EM structures of the inactive state (*Figure 5—figure supplement 1A,B*). Similarly, the recently reported conformation CaSR$^{lcc}$ has the similar B–C helix angle as CaSR$^{agonist+PAM}$, but adopts the closed–closed VFT conformation, which is a feature of the active state. It is likely that both CaSR-ECD$^{loo}$ and CaSR$^{lcc}$ are different intermediate states during the activation of CaSR. We believe that the inhibition with NB-2D11 pushes the CaSR to a completely inactive state.

At present, the prevailing view is that the principal agonist of CaSR is extracellular Ca$^{2+}$ (*Hofer and Brown, 2003*). L-amino acids, such as L-Trp, can enhance the sensitivity of CaSR toward Ca$^{2+}$ ions (*Conigrave et al., 2000*), and are considered as PAMs of the receptor (*Saidak et al., 2009*). In line with this view, a recently reported FRET study showed that Ca$^{2+}$ ions are sufficient to activate CaSR in the absence of L-amino acids, such that Ca$^{2+}$ could be considered as an agonist of CaSR, whereas L-amino acids are pure PAMs of CaSR (*Liu et al., 2020*). However, it is interesting to note that L-amino acids or their analogs are endogenous agonists of other class C GPCRs, suggesting inconsistency from the perspective of GPCR classification and evolution.

In contrast, some studies have also shown that if Ca$^{2+}$ concentration is higher than the threshold of 0.5 mM, L-amino acid can activate the receptor (*Conigrave et al., 2004*; *Conigrave et al., 2000*; *Rey et al., 2005*; *Young and Rozengurt, 2002*), indicating that Ca$^{2+}$ and L-amino acid can act as co-agonists of the receptor. Using single-cell intracellular Ca$^{2+}$ microfluorimetry, L-Trp has been shown to directly stimulate intracellular Ca$^{2+}$ mobilization in cells stably expressing CaSR, with its efficacy and potency increase with increases in concentration of Ca$^{2+}$ ions, hence providing direct evidence that L-amino acids are agonists of CaSR (*Geng et al., 2016*). However, this view has yet to be widely accepted because it is difficult to observe L-amino acids directly activating CaSR. The mode of action of Ca$^{2+}$ ion and L-amino acids on CaSR remains controversial.

Our CaSR$^{agonist+PAM}$ structure shows that TNCA and a newly identified Ca$^{2+}$ ion bind at the inter-domain cleft of the VFT module, and they both interact with both LB1 and LB2 domains to facilitate ECD closure (*Figure 3A*), therefore forming the closed state of the ligand-binding domain required for CaSR activation. This indicates that both TNCA and the Ca$^{2+}$ ion contribute to the activation of CaSR. TNCA and the bound Ca$^{2+}$ share three common binding residues S170 and D190 of LB1 domain

and E297 from LB2 domain (*Figure 3E,F*), which suggest that the CaSR is synergistically activated by TNCA and Ca$^{2+}$ ions.

Our experiments have shown that TNCA directly stimulate intracellular Ca$^{2+}$ mobilization in cells stably expressing CaSR (*Figure 3G*), suggesting that TNCA are agonists of CaSR. The structure of CaSR shows that TNCA binds at the cleft between LB1 and LB2 domains, which is the binding site for all class C GPCR agonists (*Geng et al., 2013*; *Geng et al., 2016*; *Kunishima et al., 2000*; *Muto et al., 2007*; *Tsuchiya et al., 2002*), and TNCA has a binding pattern similar to that of the endogenous agonists of mGluR and GABA$_B$ receptors (*Figure 3D*; *Geng et al., 2016*). The key coordination residues are very conserved, such as S147 and S170 (*Geng et al., 2013*; *Geng et al., 2016*; *Kunishima et al., 2000*; *Zhang et al., 2016*). Moreover, TNCA (or L-Trp) interact with residues from LB1 and LB2 to stabilize the closure of VFT, and the signal mutation of the contacting residues (T145I, S147A, S170A, Y218S, E297K) substantially reduce the function of the receptor, even if some of these residues (S147A, T145I, and Y218S) are not related to the coordinating residues of Ca$^{2+}$, hence indicating that TNCA or L-Trp plays a key role in the activation of CaSR (*Geng et al., 2016*).

In addition, Ling et al. tried to determine the cryo-EM structures of CaSR in the presence of a high concentration of Ca$^{2+}$ to address the question of whether Ca$^{2+}$ ions alone can activate CaSR in the absence of L-Trp. Three different 3D models were obtained, in which the VFT adopted closed–closed, closed–open, and open–open conformations, and an undefined L-amino acid or its derivate was buried in the closed VFT module. However, they did not obtain the closed conformation of VFT containing only Ca$^{2+}$ ion between the cleft (*Ling et al., 2021*). The results indicate that Ca$^{2+}$ ion alone is not enough to induce the closure of the VFT module even in the presence of a high concentration of Ca$^{2+}$ ion, and L-amino acid or its derivate is required to stabilize the closed conformation of VFT module. Altogether, L-amino acids are the endogenous agonists of CaSR, in agreement with that of other class C GPCRs.

It remains controversial whether Ca$^{2+}$ act alone to activate CaSR in the absence of L-amino acid. Three different groups prepared CaSR samples without L-amino acids or derivatives for crystal or cryo-EM structural studies, but they all unexpectedly obtained the active structure of CaSR or CaSR ECD with closed–closed VFT conformation containing undefined ligands (*Geng et al., 2016*; *Ling et al., 2021*; *Zhang et al., 2016*). This ligand was subsequently identified as TNCA, which had a high affinity for CaSR, and potentiated Ca$^{2+}$ activity (*Zhang et al., 2016*). As we all know, it takes a long time to purify CaSR in TNCA-free buffer for structural study; nevertheless, the endogenous TNCA would still bind to CaSR. This indicates that TNCA tightly binds to CaSR or that it is buried in closed VFT module such that it is difficult to be washed off.

Gentle washing is impossible to remove TNCA bound to CaSR in various function assays. In the absence of TNCA (or L-amino acid) and despite well-controlled assays, it is possible that endogenous TNCA would still bind to CaSR to stabilize the closed conformation of VFT. It would then appear that Ca$^{2+}$ ions directly activate the CaSR alone as TNCA remains undetected.

In addition to the role of Ca$^{2+}$ ion binding at the cleft of VFT module to stabilize the closed conformation of VFT, another role of the Ca$^{2+}$ ions should be considered, in which Ca$^{2+}$ ion is coordinated by D234, E231, and G557, and bridges the LB2 domain of one subunit and the CR domain of the second subunit. Ca$^{2+}$ ion facilitates the formation of the active conformation of CaSR, which explain why L-amino acids (or TNCA) can activate CaSR in the presence of Ca$^{2+}$ ion above the threshold concentration of 0.5 mM. We speculate that as endogenous TNCA exists, binds to CaSR with high affinity, and is difficult to be replaced by other added L-amino acids, it is challenging to observe whether L-amino acids can directly activate CaSR. Alternatively, the observed allosteric regulation is also a comprehensive result when we performed function assay. It needs the experience of the kinetics and dynamics of L-amino acid or TNCA binding with CaSR to confirm.

We observed that the LB2 domains approach each other during CaSR activation. Although we do not have a fully active conformation of the CaSR without agonist binding as evidence, crystal structures of a fully closed VFT modules of mGluR1 with or without agonist binding were previously reported (*Kunishima et al., 2000*) and have demonstrated that the proximity of both LB2 domains is an automatic process rather than an agonist-driven one. Here, we showed that NB-2D11 inhibited CaSR activation by blocking the proximity of both LB2 domains.

We analyzed how the closure of ligand bound VFT module is relayed to the signaling of 7TMDs through the CRDs. First, the rotation of LB2 domain is propagated to the large-scale transition of

intersubunit 7TMDs, which leads to rearrangement of 7TMDs interface from TM5–TM6-plane/TM5–TM6-plane interface to TM6–TM6-mediated interface. Liu et al. used FRET sensor to investigate the 7TM interface rearrangement during the activation of the CaSR through a disulfide cross-linking approach. They observed a TM4–TM5-plane/TM4–TM5-plane interface in the inactive state, but a TM6–TM6 contact in the active state of the CaSR dimer (*Liu et al., 2020*). The conformational heterogeneity of the interface of 7TMDs in the inactive state indicates that there is a possible dynamic equilibrium between the TM4–TM5 and TM5–TM6 interfaces. TM6–TM6 contact in the active state is considered to be a hallmark of class C GPCR activation. For mGluR5 and GABA$_B$ receptors, the 7TMDs rearrange from TM5–TM5 interface in the inactive state to TM6–TM6 interface in the active state (*Kim et al., 2020*; *Koehl et al., 2019*; *Mao et al., 2020*; *Papasergi-Scott et al., 2020*; *Park et al., 2020*; *Shaye et al., 2020*). In addition, structures of GABA$_B$ receptor revealed some cholesterol molecules at the interface of 7TMDs (*Kim et al., 2020*; *Mao et al., 2020*; *Papasergi-Scott et al., 2020*; *Park et al., 2020*; *Shaye et al., 2020*). It is possible that the undefined maps between the 7TMD in our full inactive structure could be sterols that separate the dimer plane–plane interface and stabilize the inactive state (*Figure 7—figure supplement 2A*). In this study, we found that the conformation of ECL2 changed from the inactive to agonist+PAM state. However, the alignment of individual 7TM of both inactive and agonist+PAM states shows that the helices are well superposed, indicating that the change of ECL2 conformation is unable to drive the rearrangements of TM4 and TM5 helices and stabilize the active conformation in the TMDs (*Koehl et al., 2019*) This observation is consistent with findings that the active state of mGluR5 is stabilized by G proteins (*Koehl et al., 2019*; *Manglik et al., 2015*; *Rosenbaum et al., 2011*). It is required to determine the structure of G protein coupling CaSR to clarify the configuration of the 7TMD in the active state.

## Materials and methods

### Key resources table

| Reagent type (species) or resource | Designation | Source or reference | Identifiers | Additional information |
|---|---|---|---|---|
| Gene (*Homo sapiens*) | CaSR | NCBI | NM_000388.4 | |
| Strain, strain background (*Escherichia coli*) | BL21(DE3) | New England Biolabs | Cat#: C2527I | *E. coli* strain for expression of the nanobody |
| Strain, strain background (*Escherichia coli*) | TG1 | Lucigen | Cat#: 60,502 | Electrocompetent cells |
| Strain, strain background (*Escherichia coli*) | TOP10F' | Huayueyang Biotech | WXR15-100S | *E. coli* strain for expression of the nanobody |
| Cell line (*Homo sapiens*) | (HEK) 293 S GnTI⁻ cells | ATCC | Cat# CRL-3022 RRID: CVCL_A785 | Mycoplasma negative |
| Cell line (*Homo sapiens*) | HEK 293T/17 | ATCC | Cat# CRL-11268 RRID: CVCL_1926 | Mycoplasma negative |
| Antibody | HA-Tag, (Mouse monoclonal) | Yeasen | 30,701ES60 | Dilution: (1/2000) |
| Antibody | Flag-Tag (DYKDDDDK), (Mouse monoclonal) | Yeasen | 30,503ES20 | Dilution: (1/2000) |
| antibody | Peroxidase AffiniPure Goat Anti-Mouse IgG (H + L) (Goat monoclonal) | Yeasen | 33,201ES60 | Dilution: (1/2500) |
| Recombinant DNA reagent | AxyPrep Plasmid Miniprep Kit | CORNING LIFE SCIENCES | Cat#:220 | |
| Recombinant DNA reagent | pMECS vector | BioVector NTCC | pMECS | Phage display vector |
| Recombinant DNA reagent | pEG BacMam vector | Addgene | Cat#:160,451 | Vector |
| Recombinant DNA reagent | pCMV-HA | Addgene | Cat#:631,604 | Vector |

*Continued on next page*

*Continued*

| Reagent type (species) or resource | Designation | Source or reference | Identifiers | Additional information |
|---|---|---|---|---|
| Recombinant DNA reagent | pcDNA3.1 | Addgene | Cat#:128,034 | Vector |
| Peptide, recombinant protein | Flag peptide | Genscript | | DYKDDDDK |
| Peptide, recombinant protein | NB88 (camel nanobody) | This study | | Isolated from phage display library of immunized cammel with hCaSR |
| Peptide, recombinant protein | NB-2D11 (camel nanobody) | This study | | Isolated from phage display library of immunized cammel with hCaSR |
| Commercial assay or kit | Luciferase assay kit | Promega | E152A | For signaling assay |
| Commercial assay or kit | SuperSignal ELISA Femto Substrate | Thermo Scientific | Cat#: 37,075 | Protein Assays and Analysis |
| Commercial assay or kit | Fluo-4, AM, Cell Permeant | YEASEN | 40,704ES50 | |
| Chemical compound, drug | TNCA | aladdin | 42438-90-4 | |
| Chemical compound, drug | NPS-2143 | aladdin | 284035-33-2 | |
| Chemical compound, drug | cinacalcet | aladdin | 364782-34-3 | |
| Chemical compound, drug | Lauryl Maltose Neopentyl Glycol (LMNG) | Anatrace | NG310 | Membrane protein purification |
| Chemical compound, drug | Glyco-Diosgenin (GDN) | Anatrace | GDN101 | Membrane protein purification |
| Chemical compound, drug | Cholesterol Hemisuccinate tris Salt (CHS) | Anatrace | CH210 | Membrane protein purification |
| Chemical compound, drug | TMB substrate | Thermo Fisher Scientific | 34,021 | Protein Assays and Analysis |
| Software, algorithm | cryoSPARC | https://cryosparc.com | Version 3.0.0 RRID:SCR_016501 | Cryo-EM data processing |
| Software, algorithm | PHENIX | http://www.phenix-online.org/ | Version 1.19.2 RRID:SCR_014224 | Structure refinement |
| Software, algorithm | Coot | Coot (cam.ac.uk) | Version 0.9.4 RRID:SCR_014222 | Structure refinement |
| Software, algorithm | MolProbity | DOI:10.1107/S0907444909042073 | RRID:SCR_014226 | Structure verification |
| Software, algorithm | UCSF Chimera | https://wwwcgl.ucsf.edu/chimera/ (PMID:15264254) | Version 1.15 RRID:SCR_004097 | Initial homology model docking |
| Software, algorithm | PyMol | Schrodinger | Version 2.5 RRID:SCR_000305 | Structural visualization/figure preparation |
| Software, algorithm | GraphPad Prism 7 | GraphPad | RRID:SCR_002798 | Analysis of signaling data |
| Other | Lipofectamine 2000 | Invitrogen | 11668030 | Transfection reagent for signaling assay |

## Cell lines

(HEK) 293 S GnTI⁻ cells (human) were purchased from ATCC (Cat# CRL-3022 RRID:CVCL_A785), which were grown in FreeStyle 293 medium (Gibco) supplemented with 2 % (v/v) FBS (Gibco) and 8 % $CO_2$ for maintenance. HEK293T/17 cells (ATCC, Cat# CRL-11268 RRID:CVCL_1926) were grown in Dulbecco's modified eagle medium (DMEM, Gibco) supplemented with 10 % (v/v) FBS and 5 % $CO_2$. All cell lines were grown at 37°C. All the cell lines tested negative for mycoplasma contamination.

## Nanobody library generation

Camel immunizations and nanobody library generation were performed as described previously (*Pardon et al., 2014*). Animal work was conducted under the supervision of Shanghai Institute of

Materia Medica, Chinese Academy of Sciences. In brief, two camels were immunized subcutaneously with approximately 1 mg human CaSR protein combined with equal volume of Gerbu FAMA adjuvant once a week for seven consecutive weeks. Three days after the last immunization, peripheral blood lymphocytes (PBLs) were isolated from the whole blood using Ficoll-Paque Plus according to manufacturer's instructions. Total RNA from the PBLs was extracted and reverse transcribed into cDNA using a Super-Script III FIRST-Strand SUPERMIX Kit (Invitrogen). The VHH encoding sequences were amplified with two-step enriched-nested PCR using VHH-specific primers and cloned between *PstI* and *BsteII* sites of pMECS vector. Electro-competent *E. coli* TG1 cells (Lucigen) were transformed and the size of the constructed nanobody library was evaluated by counting the number of bacterial colonies. Colonies were harvested and stored at −80 °C.

## Nanobody identification by phage display

*E. coli* TG1 cells containing the VHH library were superinfected with M13KO7 helper phages to obtain a library of VHH-presenting phages. Phages presenting CaSR-specific VHHs were enriched after three rounds of biopanning. For each panning round, phages were dispensed into CaSR coated 96 wells (F96 Maxisorp, Nunc), incubated for 2 hr on a vibrating platform (700 r.p.m), and subsequently washed 10 times with PBST and five times with PBS. The retained phages were eluted with 0.25 mg ml$^{-1}$ trypsin (Sigma-Aldrich). The collected phages were subsequently amplified in *E. coli* TG1 cell for consecutive rounds of panning. After the third rounds of biopanning, 200 positive clones were picked and infected with M13KO7 helper phages to obtain the VHH-presenting phages.

## ELISA to select CaSR VHHs

The wells of ELISA plates were coated with 2 μg ml$^{-1}$ neutravidin in PBS overnight at 4 °C. Biotinylated CaSR (2 μg ml$^{-1}$) was added into each well. Then the wells were blocked with 5 mg ml$^{-1}$ non-fat milk powder in PBS. One hundred microliter supernatant of HA-tagged CaSR VHH was added into each well with 1 hr incubation at 4 °C, followed by incubation with horseradish peroxidase (HRP)-conjugated anti-HA (Yeasen). TMB substrate (Thermo Fisher Scientific) was added, and the reactions were stopped by 2 M H$_2$SO$_4$. Measurement was performed at 450 nm.

## Purification of NB-2D11

NB-2D11 was cloned into a pMECS vector (NTCC) that contains a PelB signal peptide and a hemagglutinin (HA) tag followed by a 6× histidine tag at the C-terminus. It was expressed in the periplasm of *E. coli* strain TOP10F' (Huayueyang Biotech) and grown to a density of OD$_{600nm}$ 0.6–0.8 at 37 °C in 2YT media containing 100 μg/ml Ampicillin, 0.1 % (w/v) glucose and 1 mM MgCl$_2$, and then induced with 1 mM IPTG at 28 °C for 12 hr. The bacteria were harvested by centrifugation and resuspended in a buffer containing 20 mM HEPES pH 7.5, 150 mM NaCl, 1 mM PMSF, and lysed by sonication, then centrifuged at 4000 r.p.m. to remove cell debris. The supernatant was loaded onto Ni-NTA resin and further eluted in elution buffer containing 20 mM HEPES pH 7.5, 150 mM NaCl, and 300 mM imidazole. The elution was purified by gel filtration chromatography using a HiLoad 16/600 Superdex 75 pg column in 150 mM NaCl with 20 mM HEPES pH7.5. Finally, NB-2D11 was flash-frozen in liquid nitrogen until further use.

## Purification of inactive state CaSR bound to NPS-2143 and NB-2D11

Human CaSR (1-870) followed by a Flag epitope tag (DYKDDDD) at the C-terminus was cloned into a modified pEG BacMam vector (*Goehring et al., 2014*) for expression in baculovirus-infected mammalian cells. Human embryonic kidney (HEK) 293 GnTI⁻ cells (ATCC) were infected with baculovirus at a density of 2.5 × 10$^6$ cells per ml at 37 °C in 8 % CO$_2$. Ten millimolar  sodium butyrate was added 12–16 hr postinfection, then cells were grown for 48 hr at 30 °C with gentle rotation.

The infected cells were harvested by centrifugation at 4000 g for 30 min, resuspended, and homogenized using a dounce tissue grinder (WHEATON) in hypotonic buffer (20 mM HEPES pH7.5, 10 mM NaCl, 1 mM CaCl$_2$, 10 % glycerol, 1× cocktail of protease inhibitor, and 1 μM NPS-2143). Cell membrane was collected by ultra-centrifugation at 40,000 r.p.m. in a Ti-45 rotor (Beckman Coulter) for 1 hr. Then the membrane was resuspended and solubilized in buffer containing 20 mM HEPES, 150 mM NaCl, 1 mM CaCl$_2$, 10 % glycerol, 1 μM NPS-2143, 1 % (w/v) lauryl maltose neopentyl glycol (LMNG) (Anatrace), and 0.1 % (w/v) cholesteryl hemisuccinate TRIS salt (CHS) (Anatrace) for 1 hr at

4 °C with constant stirring. The supernatant was collected by ultra-centrifugation at 40,000 r.p.m. for 1 hr and applied to an anti-Flag M2 antibody affinity column (Sigma-Aldrich). After receptor binding to the M2 column, the resin was washed with 20 mM HEPES, 150 mM NaCl, 1 mM $CaCl_2$, 10 % glycerol, 1 µM NPS-2143, 0.1 % LMNG, 0.01 % CHS. The column was washed stepwise with decreasing proportion of LMNG and increasing concentration of GDN/CHS to 0.2%/0.02 %. CaSR was then eluted with 20 mM HEPES, 150 mM NaCl, 1 mM $CaCl_2$, 10 % glycerol, 1 µM NPS-2143, 0.02 % GDN, 0.002 % CHS, and 0.2 mg $ml^{-1}$ Flag peptide.

CaSR was further purified by ion-exchange chromatography using a Mono Q 5/50 GL column. Peak fractions were assembled and incubated with a 1.2 molar excess of NB-2D11 for 1 hr before injection on a Superose 6 Increase 10/300 GL column. Fractions of CaSR-NB-2D11 complex in buffer containing 20 mM HEPES, 150 mM NaCl, 1 mM $CaCl_2$, 1 µM NPS-2143, 0.002 % GDN, and 0.0002 % CHS were pooled and concentrated to approximately 5 mg $ml^{-1}$ for further cryo-EM sample preparation.

## Purification of active state CaSR bound to cinacalcet and TNCA

Infected cells (described above) were collected and resuspended in hypotonic buffer (20 mM HEPES pH 7.5, 10 mM NaCl, 10 mM $CaCl_2$, 10 % glycerol, 1× cocktail of protease inhibitor, 1 µM cinacalcet, and 1 µM TNCA). Cell membrane was collected by ultra-centrifugation at 40,000 r.p.m. for 1 hr, resuspended, and solubilized in buffer containing 20 mM HEPES, 150 mM NaCl, 10 mM $CaCl_2$, 10 % glycerol, 1 µM cinacalcet, 1 µM TNCA, 1 % LMNG, and 0.1 % CHS for 1 hr at 4 °C. The supernatant was collected by ultra-centrifugation and applied to an anti-Flag M2 antibody affinity column. After receptor binding to the M2 column, the resin was washed with 20 mM HEPES, 150 mM NaCl, 10 mM $CaCl_2$, 10 % glycerol, 1 µM cinacalcet, 1 µM TNCA, 0.1 % LMNG, 0.01 % CHS. LMNG was exchanged for GDN to a proportion of 0.2 % in stepwise washing. CaSR was then eluted with 20 mM HEPES, 150 mM NaCl, 10 mM $CaCl_2$, 10 % glycerol, 1 µM cinacalcet, 1 µM TNCA, 0.02 % GDN, 0.002 % CHS, and 0.2 mg $ml^{-1}$ Flag peptide.

CaSR was further purified by Mono Q 5/50 GL column. Peak fractions were assembled and injected to a Superose 6 Increase 10/300 GL column. Fractions of CaSR in buffer containing 20 mM HEPES, 150 mM NaCl, 10 mM $CaCl_2$, 1 µM cinacalcet, 1 µM TNCA, 0.002 % GDN, and 0.0002 % CHS were pooled and concentrated to approximately 5 mg $ml^{-1}$ for further cryo-EM sample preparation.

## Cryo-EM sample preparation and data acquisition

Three microliters of inactive or active CaSR protein was applied to glow-discharged holey carbon 300 mesh grids (Quantifoil Au R1.2/1.3, Quantifoil MicroTools), respectively. The grids were blotted for 2 s and flash-frozen in liquid ethane using a Vitrobot Mark IV (Thermo Fisher Scientific) at 4 °C and 100 % humidity. Cryo-EM data was collected on a Titan Krios microscope (Thermo Fisher Scientific) at 300 kV accelerating voltage equipped with a Gatan K3 Summit direct election detector at a nominal magnification of 81,000× in counting mode at a pixel size of 1.071 Å. Each micrograph contains 36 movie frames with a total accumulated dose of 70 electrons per Å. The defocus range was set –1.5 to –2.5 µm. A total of 5706 and 4981 movies for active and inactive CaSR were collected for further data processing, respectively.

## Data processing and 3D reconstruction

All images were aligned and summed using MotionCor2 (*Zheng et al., 2017*). Unless otherwise specified, single-particle analysis was mainly executed in RELION 3.1 (*Zivanov et al., 2020*). After CTF parameter determination using CTFFIND4 (*Rohou and Grigorieff, 2015*), particle auto-picking, manual particle checking, and reference-free 2D classification, 1,546,992 and 2,208,402 particles remained in the active and inactive datasets, respectively. The particles were extracted on a binned dataset with a pixel size of 4.42 Å and subjected to 3D classification, with the initial model generated by ab-initio reconstruction in cryoSPARC (*Punjani et al., 2017*).

For the CaSR active state dataset, 3D classification resulted in extraction of 36.6 % good particles with a pixel size of 1.071 Å. The particles were subsequently subjected to an auto-refine procedure, yielding a 4.3-Å-resolution map. Afterwards, particles were polished, sorted by carrying out multiple rounds of 3D classifications, yielding a dataset with 560,366 particles, generating a 3.3-Å-resolution map. Another round of 3D classification focusing the alignment on the complex, resulted in two conformations with high-quality features. After refinement, the resolution levels of these two

maps improved to 3.43 Å and 2.99 Å. Particle subtractions on the ECD and TM domains were also performed to further improve the map quality. After several rounds of 3D classifications, ECD map has a resolution of 3.07 Å with 493,869 particles, while that for TM is 4.3 Å with 389,105 particles.

For the CaSR inactive state dataset, 3D classification resulted in extraction of 55 % good particles with a pixel size of 1.071 Å. The particles were subsequently subjected to an auto-refine procedure, yielding a 6.0-Å-resolution map. Afterwards, particles were further sorted with another round of 3D classification focusing the alignment on the TM domain, resulted in 37.7 % particles with high-quality features. Further 3D classification on the whole complex separates three different orientations of ECD relative to TM domain. After refinement, the resolution levels of these three maps improved to 5.79 Å, 6.88 Å, and 7.11 Å. Particle subtractions on the ECD and TM domains were also performed to further improve the map quality. After several rounds of 3D classifications, ECD map has a resolution of 4.5 Å with 253,294 particles, while that for TM is 4.8 Å with 691,246 particles.

## Model building and refinement

The crystal structures of CaSR ECD in apo and active forms (PDB Code: 5K5T, 5K5S) were used as initial templates for the ECD of the CaSR. The cryo-EM structures of mGluR5 in resting and active forms (PDB Code: 6N52, 6N51) were used as initial models for the TM domains of the receptor. The agonist TNCA was generated by COOT (*Emsley and Cowtan, 2004*) and PHENIX.eLBOW (*Adams et al., 2010*). The initial templates of ECDs and TMDs were docked into the cryo-EM maps of CaSR using UCSF Chimera (*Goddard et al., 2018*) to build the initial models of CaSR in inactive and active forms. Then the main chains and side chains of the initial models were manually rebuilt in COOT. The models were subsequently performed by real-time refinement in PHENIX.

## Intracellular $Ca^{2+}$ flux assay

HEK293T cells (ATCC) were transiently transfected with wild-type or mutant full-length CaSR plasmids. Five micrograms DNA plasmid was incubated with 15 µl lipofectimin in 500 µl OptiMEM for 10 min at room temperature and then added to the cells for overnight incubation at 37 °C. The transfected cells were trypsinized and seeded in 96-well plates. On the day of assay, the cells were incubated with loading medium containing 20 mM HEPES, 125 mM NaCl, 4 mM KCl, 1.25 mM $CaCl_2$, 1 mM $MgSO_4$, 1 mM $Na_2HPO_4$, 0.1% D-glucose, and 0.1% BSA at 37°C for 4 hr. Then the buffer was replaced with 100 µl of buffer containing Fluo-4 at 37°C for 1 hr incubation, and then placed into the FLIPR Tetra High Throughput Cellular Screening System. Data was analyzed by non-linear regression in Prism (GraphPad Software). Data points represent average ± SEM of quadruplicate measurements.

## Surface plasmon resonance

SPR experiments were performed using a Biacore T200 instrument (GE Healthcare). The system was flushed with running buffer (20 mM HEPES pH 7.4, 150 mM NaCl, 0.05 % Tween 20), and all steps were performed at 25 °C chip temperature. The CaSR ECD flowed through the negatively charged chip at a concentration of 1 mg/ml and a flow rate of 10 µl/min for 1 min and was captured by amino-carboxyl coupling reaction. It was followed by nanobody NB-2D11 that went through the chip at a series of concentration (30 µl/min, association: 90 s, dissociation: 220 s). All Biacore kinetic experiment data were obtained using Biacore S200 Evaluation Software to calculate the $K_D$, which is the ratio of $k$d/$k$a.

## ELISA for cell-surface expression

ELISA was performed as a control to quantify cell surface expression of each CaSR mutant (*Mos et al., 2019*). In brief, HEK293T cells were transiently transfected with wild-type (WT) or mutant full-length CaSR plasmids. Five micrograms DNA plasmid was incubated with 15 µl lipofectimine (Invitrogen) in 500 µl OptiMEM (Gibco) for 10 min at room temperature and then added to the cells for overnight incubation at 37°C. The transfected cells were trypsinized and seeded in poly-D-lysine-coated 96-well plates (Greiner bio-one, cat# 655083). On the day of assay, cells were fixed with 4 % paraformaldehyde in PBS for 20 min and washed twice. The cells were incubated with blocking buffer containing 3 % skim milk in PBS followed by incubation for 1 hr with anti-Flag antibody (Yeasen) in blocking buffer. The cells were then incubated with horseradish peroxidase goat anti-mouse IgG (Yeasen) diluted 1:5000 in blocking solution for 1 hr. Chemiluminescence was measured on a Tecan plate reader immediately

after addition of 10 µl/well SuperSignal ELISA Femto Substrate (Thermo Fisher Scientific). The results show that each CaSR mutant displays similar fluorescence intensity as that of wild type, which indicates that the elimination of the calcium response is not caused by misfolding or mis-trafficking of the receptor. All mutants were well-expressed on the cell surface compared to the WT receptor.

## Acknowledgements

The cryo-EM data were collected at the Cryo-Electron Microscopy Research Center, Shanghai Institute of Materia Medica, Chinese Academy of Sciences. This work is supported by National Natural Science Foundation of China (No. 31670743), Strategic Priority Research Program of the Chinese Academy of Sciences (No. XDA12040326), Science and Technology Commission of Shanghai Municipality (No. 18JC1415400), Joint Research Fund for Overseas, Hong Kong and Macao Scholars (No. 81628013), Natural Science Foundation of Shanghai (16ZR1442900), National Science Foundation for Young Scholar projects (118180359901), and grants from Shanghai Institute of Materia Medica, Chinese Academy of Sciences (CASIMM0120164013, SIMM1606YZZ-06, SIMM1601KF-06, 55201631121116101, 55201631121108000, 5112345601, 2015123456005).

## Additional information

### Funding

| Funder | Grant reference number | Author |
|---|---|---|
| National Natural Science Foundation of China | No. 31670743 | Yong Geng |
| Science and Technology Commission of Shanghai Municipality | No. 18JC1415400 | Yong Geng |
| Joint Research Fund for Overseas Chinese Scholars and Scholars in Hong Kong and Macao | No. 81628013 | Yong Geng |
| Natural Science Foundation of Shanghai | 16ZR1442900 | Yong Geng |
| Shanghai Institute of Materia Medica, Chinese Academy of Sciences | CASIMM0120164013 | Yong Geng |
| Shanghai Institute of Materia Medica, Chinese Academy of Sciences | SIMM1606YZZ-06 | Yong Geng |
| Shanghai Institute of Materia Medica, Chinese Academy of Sciences | SIMM1601KF-06 | Yong Geng |
| Shanghai Institute of Materia Medica, Chinese Academy of Sciences | 55201631121116101 | Yong Geng |
| Shanghai Institute of Materia Medica, Chinese Academy of Sciences | 55201631121108000 | Yong Geng |
| Shanghai Institute of Materia Medica, Chinese Academy of Sciences | 5112345601 | Yong Geng |
| Shanghai Institute of Materia Medica, Chinese Academy of Sciences | 2015123456005 | Yong Geng |

| Funder | Grant reference number | Author |
|---|---|---|
| National Natural Science Foundation of China | 118180359901 | Yong Geng |

The funders had no role in study design, data collection and interpretation, or the decision to submit the work for publication.

## Author contributions

Xiaochen Chen, Data curation, Formal analysis, Investigation, Methodology, Project administration, Validation, Visualization, Writing – original draft, Writing – review and editing; Lu Wang, Data curation, Investigation, Validation, Writing – original draft, Writing – review and editing; Qianqian Cui, Investigation, Writing – review and editing; Zhanyu Ding, Data curation, Investigation, Validation, Writing – original draft; Li Han, Yongjun Kou, Wenqing Zhang, Haonan Wang, Xiaomin Jia, Mei Dai, Zhenzhong Shi, Yuying Li, Xiyang Li, Investigation; Yong Geng, Conceptualization, Data curation, Formal analysis, Funding acquisition, Investigation, Methodology, Project administration, Resources, Supervision, Validation, Visualization, Writing – original draft

## Author ORCIDs

Xiaochen Chen (iD) http://orcid.org/0000-0002-0426-9547
Lu Wang (iD) http://orcid.org/0000-0001-6336-5806
Qianqian Cui (iD) http://orcid.org/0000-0002-5962-9298
Zhanyu Ding (iD) http://orcid.org/0000-0002-8136-2243
Yong Geng (iD) http://orcid.org/0000-0001-7144-3878

## Ethics

The animal work was approved and under the supervision of Shanghai Institute of Materia Medica, Chinese Academy of Sciences (Permit Number: SYXK 2015-0027).

## Decision letter and Author response

Decision letter https://doi.org/10.7554/eLife.68578.sa1
Author response https://doi.org/10.7554/eLife.68578.sa2

# Additional files

## Supplementary files

• Transparent reporting form

## Data availability

All data is available in the main text or the supplementary materials. Cryo-EM maps of active CaSR in complex with TNCA and inactive CaSR in complex with NB-2D11 have been deposited in the Electron Microscopy Data Bank under accession codes: EMD-30997 (NB-2D11 bound CaSR), EMD-30996 (TNCA bound CaSR). Atomic coordinates for the CaSR in complex with TNCA or NB-2D11 have been deposited in the Protein Data Bank under accession codes: 7E6U (NB-2D11 bound CaSR), 7E6T (TNCA bound CaSR).

The following dataset was generated:

| Author(s) | Year | Dataset title | Dataset URL | Database and Identifier |
|---|---|---|---|---|
| Geng Y, Chen XC | 2021 | Cryo-EM structure of CaSR in complex with NB-2D11 | http://www.rcsb.org/structure/unreleased/7E6U | RCSB Protein Data Bank, 7E6U |
| Chen X, Wang L, Ding Z, Cui Q, Han L, Kou Y, Zhang W, Wang H, Jia X, Dai M, Shi Z, Li Y, Li X, Geng Y | 2021 | Cryo-EM structure of CaSR in complex with TNCA | http://www.rcsb.org/structure/unreleased/7E6T | RCSB Protein Data Bank, 7E6T |

The following previously published datasets were used:

| Author(s) | Year | Dataset title | Dataset URL | Database and Identifier |
|---|---|---|---|---|
| Geng Y, Mosyak L, Kurinov I, Zuo H, Sturchler E, Cheng TC, Subramanyam P, Brown AP, Brennan SC, Mun H-C, Bush M, Chen Y, Nguyen T, Cao B, Chang D, Quick M, Conigrave A, Colecraft HM, McDonald P, Fan QR | 2016 | Crystal structure of the inactive form of human calcium-sensing receptor extracellular domain | https://www.rcsb.org/structure/5K5T | RCSB Protein Data Bank, 5K5T |
| Geng Y, Mosyak L, Kurinov I, Zuo H, Sturchler E, Cheng TC, Subramanyam P, Brown AP, Brennan SC, Mun H-C, Bush M, Chen Y, Nguyen T, Cao B, Chang D, Quick M, Conigrave A, Colecraft HM, McDonald P, Fan QR | 2016 | Crystal structure of the active form of human calcium-sensing receptor extracellular domain | https://www.rcsb.org/structure/5K5S | RCSB Protein Data Bank, 5K5S |
| Koehl A, Hu H, Feng D, Sun B, Weis WI, Skiniotis GS, Mathiesen JM, Kobilka BK | 2019 | Metabotropic Glutamate Receptor 5 bound to L-quisqualate and Nb43 | https://www.rcsb.org/structure/6N51 | RCSB Protein Data Bank, 6N51 |
| Mao C, Shen C, Li C, Shen D, Xu C, Zhang S, Zhou R, Shen Q, Chen L, Jiang Z, Liu J, Zhang Y | 2020 | Cryo-EM structure of the baclofen/BHFF-bound human GABA(B) receptor in active state | https://www.rcsb.org/structure/7C7Q | RCSB Protein Data Bank, 7C7Q |

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
