## [Decision Letter]

**Acceptance summary:**

This manuscript describes the cryo-EM structures of the human calcium-sensing receptor CaSR. Through the use of both agonists and negative allosteric modulators, these structures reveal conformational changes of the receptor that contribute to signaling. CaSR has unquestionable medical relevance, and the topic is of interest to structural biologists and in cell signaling.

**Decision letter after peer review:**

Thank you for submitting your article "Structural insights into the activation of human calcium- sensing receptor" for consideration by *eLife*. Your article has been reviewed by 3 peer reviewers, including Randy B Stockbridge as Reviewing Editor and Reviewer #1, and the evaluation has been overseen by Kenton Swartz as the Senior Editor. The following individual involved in review of your submission has agreed to reveal their identity: Jean-Philippe Pin (Reviewer #2).

Essential revisions:

1. For the mutant analysis, it is essential to include a control for cell surface expression. Many of the mutants eliminate the calcium response, but this may be an indirect effect if the mutant causes misfolding of mis-trafficking of the receptor.

2. Additional quality metrics for the cryo-EM data and models should be reported. These include more extensive maps (for example, maps around excised regions of secondary structure to assess overall correspondence of maps and models, and maps of the sidechains in the ligand binding site), FSC curve and the distribution of viewing angles of the particles used in the reconstructions, and an analysis of model geometry. Please ensure that Figure 1—figure supplement 2 actually shows maps, and not a surface rendering.

3. The authors should revise the manuscript to engage more with the current literature on CaSR. Doing so is essential to support some of the structural observations. In particular:

– the authors should compare the current structures of CaSR with the recent structures of Ling et al., 2021 for the same receptor. The authors should elaborate on their rationale for arguing that the current structures are more representative of the inactive state than prior models. The authors should explain why the prior model of the inactive state is described as "controversial."

– The authors propose a Calcium binding site that corresponds to one proposed to be essential for CaSR activation by Liu et al., (PNAS 2020), based on models, mutagenesis, and the use of a conformational FRET-based sensor. This work must be cited. Although such a site is very likely to bind Ca ions, the maps provided by the cryo-EM images are far from being sufficient to guarantee that the density observed corresponds to a Ca ion, rather than to another ion. The authors must clearly indicate this and argue why they think this is probably a Ca ion.

– The relative movement of the 7TM domains observed in these structures also fully agree with what was proposed by Liu et al., (2020) through a Cys cross-linking approach and functional analysis of the cross-linked dimers. This study must be discussed and cited.

4. The authors should not refer to the agonist+PAM conformation as the active one, as this is very unlikely since the 7TM domain conformation is almost identical to that observed in the nanobody stabilized inactive state. Please refer to this state as the agonist+PAM bound state.

*Reviewer #1 (Recommendations for the authors):*

1. The conclusion that the Ca^2+^ and TCA site are cooperative is based on structural evidence. Have any experiments in the literature shown cooperativity between TCA and Ca^2+^?

2. The nanobody is an important part of this story. It would be helpful to provide more information on the selection of the nanobody. Was the inactive conformation the target of the nanobody selection? Was anything done during the screening to select for a nanobody against the inactive conformation? I suggest introducing the nanobody selection and activity assays earlier in the manuscript.

3. The authors should elaborate on their rationale for arguing that the current structures are more representative of the inactive state than prior models. Why is the prior model of the inactive state described as "controversial?"

4. The quality of the model (bonds, angles, C-β deviations, etc) should be evaluated using a web server like MolProbity to ensure that the model statistics are satisfactory. These should be reported.

*Reviewer #2 (Recommendations for the authors):*

1 – The authors propose a Calcium binding site that corresponds to one proposed to be essential for CaSR activation by Liu et al., (PNAS 2020), based on models, mutagenesis, and the use of a conformational FRET-based sensor. This work must be cited. Although such a site is very likely to bind Ca ions, the maps provided by the cryo-EM images are far from being sufficient to guarantee that the density observed corresponds to a Ca ion, rather than to another ion. The authors must clearly indicate this and argue why they think this is probably a Ca ion.

2 – The authors should not refer to the agonist+PAM conformation as the active one, as this is very unlikely since the 7TM domain conformation is almost identical to that observed in the nanobody stabilized inactive state. I recommend they refer to this state as the agonist+PAM bound state.

3 – Most of the inactive structures of the isolated CaSR VFT dimers are all in an orientation corresponding to the active one of the mGlu receptors, with both lobes 2 being in close contact. The recent full length structures revealed three possible inactive states, the VFT dimer in the resting closed closed (Rcc), Resting open closed (Rco) or resting open-open (Roo). Here, the authors report a conformation corresponding to the Roo state that corresponds to the inactive state of mGluR VFTs, but they do not provide a clear demonstration that this is indeed the case. One argument that can be used is based on the work by Liu et al., (2020) that show that the CaSR receptor VFT dimer undergoes a similar change in its orientation during activation, as that of the mGlu receptors. Another argument will be the analysis of the functional effect of the identified nanobody. It is shown to inhibit the receptor activity (Figure 5b), but can a constitutive activity of the CaSR receptor resulting from the other resting conformation be detected? If so, can it be inhibited by NB-2D11?

4 – The relative movement of the 7TM domains observed in these structures also fully agree with what was proposed by Liu et al., (2020) through a Cys cross-linking approach and functional analysis of the cross-linked dimers, with dimers cross-linked through TM6 being constitutively active, while Ca effect was dramatically reduced in TM4 or 5 cross-linked dimers. This study must be discussed and cited.

5 – The authors must also be clear on the relative role of TNCA (or other L-AA) and Ca ions on the activation of the receptor. Using a well-controlled assay, Liu et al., (2020) reported that Ca ions are sufficient to activate the CaSR, even in the absence of L-AA, such that Ca ion can be considered as the agonist of the CaSR. In contrast, any tested L-AA did not activate the receptor, but largely potentiated the Ca effect, such that these L-AA should be considered as PAMs of the CaSR. This must be clarified in the text.

6 – The intermediate state presented should be better explain in the text.

7 – Please be more specific in Figure 4 on how the distances between the C-termini of the VFTs were measured. Indicate which atom was used as a reference.

*Reviewer #3 (Recommendations for the authors):*

As discussed above, I believe the manuscript could be strengthened by better discussing the distinct aspects of the structures compared to the Ling et al., published paper.

Also I would recommend the authors to go through the manuscript again and to double check typos and grammatical errors. I have noted a few here:

L9: our data shows

L33: through which can be removed.

L44: the active structure has demonstrated a calcium…I would rather say the active structure presents a calcium…

This is actually the case several times in the manuscript (L57, 65…). When speaking about structural or map features, I would suggest to replace "has demonstrated" or "have demonstrated" or 'has shown' by "present" or similar.

L78: the sentence is not clear ans should be rewritten:

and the closed conformation of the VFT module has shown in the active state, which is stabilized by the TNCA binding at the VFT module and 3 distinct calcium binding sites within each ECD

L102: throughout the manuscript you wrote promoter instead of protomer.

L231: is consisted (consistent)

L240: contacting with the map does not make sense in the sentence.

L254: presents similar, except …similar what? This sentence should be written in a clearer way.

L290: There is a common interface constitute of residues 759-763 at ECL2 and residues 601-604 at the C-terminal of CRD (Figure 7A), and the interface in the active conformation is more compact than that of in the inactive conformation

This sentence should be written in a clearer way.

L301: relay should be relays. I would however write 'ECL2 seems to play a key role in relaying…..'

L322-325: This sentence should be rewritten, as it is not clear and it contains grammatical errors (allows).

L343-344: thirdly, the closure of VFT that stabilized by the agonist and Ca^2+^ ion is a landmark event during the CaSR activation.

You should add 'is' before satbilized.

L348-349: This sentence should be written in a clearer way.

L374: same error as before with consisted instead of consistent.

---

## [Author Response]

Essential revisions:1. For the mutant analysis, it is essential to include a control for cell surface expression. Many of the mutants eliminate the calcium response, but this may be an indirect effect if the mutant causes misfolding of mis-trafficking of the receptor.

We agree with the reviewer that it is essential to measure the cell surface expression levels of the mutants to analyze their effects on the receptor function. The cell-surface expression levels of the WT and mutant CaSR constructs were examined using ELISA as described previously (Mos et al., 2019). The cell surface expression levels of the mutants of the TNCA and calcium common binding sites (S170K, D190K, E297K and Y489F) were approximately 80-100% of the wild-type level. However, each of these mutants nearly abolished or reduced ca^2+^-dependent receptor response, indicating that their effects on TNCA and calcium binding and receptor function are not only due to the decrease in cell surface expression.

The expression of each mutant of F789A and F792A from TM5 on the cell surface were higher than that of the WT, however, both mutations still significantly attenuate Ca^2+^-induced receptor activity.

In order to verify the proximity of 7TMDs during the activation, we mutated P823R to disrupt the contact of TM6. The result demonstrated that the expression of P823R mutants was similar to that of the wild type, but after the mutation, the activity of the receptor was seriously affected. Our result shows that the P823R mutant interferes with the active interface driven by TM6, resulting in a decrease in the receptor function.

The structure reveals that the interaction between W590 and K601 of CRD and D758 and E759 of ECL2 relays the conformational changes of VFT to the TM domain. The cell surface expression level of ∆758-759 mutant was comparable to that of WT, while W590E and K601E mutants were expressed on the cell surface at approximately 40-50% of WT level. Each of these mutant leads to a largely decreased effect of Ca^2+^-stimulated intracellular Ca^2+^ mobilization in cells, which indicates that the contact formed by CRD and ECL2 plays an important role in converting the ligand binding domain conformational change into the intra-7TM rearrangement.

2. Additional quality metrics for the cryo-EM data and models should be reported. These include more extensive maps (for example, maps around excised regions of secondary structure to assess overall correspondence of maps and models, and maps of the sidechains in the ligand binding site), FSC curve and the distribution of viewing angles of the particles used in the reconstructions, and an analysis of model geometry. Please ensure that Figure 1—figure supplement 2 actually shows maps, and not a surface rendering.

We thank the reviewer for pointing out our negligence. We have added more extensive maps that contain more information of the receptors, including the regions of secondary structure of the maps and models, and the side chains of the ligand binding site, FSC curve and the distribution of viewing angles. We have replaced the surface rendering with the actual map in Figure 2—figure supplement 2.

3. The authors should revise the manuscript to engage more with the current literature on CaSR. Doing so is essential to support some of the structural observations. In particular:– the authors should compare the current structures of CaSR with the recent structures of Ling et al. 2021 for the same receptor. The authors should elaborate on their rationale for arguing that the current structures are more representative of the inactive state than prior models. The authors should explain why the prior model of the inactive state is described as "controversial."

Following the reviewer’s suggestion, we have compared the recently reported structures of CaSR with our structure in the results and Discussion sections. Ling *et al.* determined structures of CaSR homodimer in distinct conformations and their findings help us understand the global and local conformational transitions during CaSR activation. They obtained three different structures of CaSR in the Ca^2+^-free sample, which were considered to represent CaSR in inactive states. Their three models showed similar architecture, but the VFT domains adopted three different conformations, including closed-closed, open-closed, and open-open conformations, designated as CaSR^Icc^, CaSR^Ioc^, and CaSR^Ioo^, respectively. They indicated that the CaSR in the inactive state had conformational heterogeneity. However, they only built the structure of CaSR in the Icc conformation due to the relatively lower resolution. They also determined the structure of the CaSR-L-Trp complex, designated as CaSR^Trp^, as the intermediate state of CaSR during activation. They compared the structures of CaSR^Trp^ and CaSR^Icc^ and found that both structures were identical. Moreover, they determined two structures of CaSR in the active state, one was bound with Ca^2+^ and L-Trp and designated as CaSR^Acc^, while another was obtained in the presence of a high concentration of Ca^2+^ ions and designated as CaSR^Ca^. Structural comparison demonstrated that the structure of CaSR^Ca^ was almost identical to that of CaSR^Acc^. Except from the structure of CaSR^Ca^ in active state, the inactive conformation of CaSR (CaSR^Ioc^ and CaSR^Ioo^) were captured in the high concentration of Ca^2+^ ions, unfortunately, they still were not resolved due to lower resolution. In conclusion, they determined two structures of CaSR (CaSR^Acc^ and CaSR^Ca^) in the active states and two structures of CaSR (CaSR^Icc^ and CaSR^Trp^) in the inactive states (or intermediate state), and also observed two inactive conformations of CaSR (CaSR^Ioc^ and CaSR^Ioo^, but their structures were not built). They reported that ca^2+^ induced a conformational transition of CaSR from CaSR^Icc^ to CaSR^Acc^, as CaSR^Icc^ and CaSR^Acc^ had striking different overall architectures, especially, the CRDs and TMDs separated in the dimeric receptor, while their VFT domains had tiny changes. They also reported that the L-Trp binding induced the closed-closed conformation of the VFT domains, leading to conformational conversion from CaSR^Ioo^ to CaSR^Icc^.

Previous structural and functional studies have suggested a universal activation mechanism for class C GPCR. First, there are two typical conformational changes for the VFT domains in the dimeric receptor from the full inactive state to the full active state. One is that the conformation of VFT domain was converted from open to closed for the change of interdomain in one protomer, another is that the B-C helix angle on the interfaces of LB1-LB1 dimer sharply rotated from inactive state to active state. Second, the closure of VFT domain brings the CRD into close proximity which leads to rearrangement of the 7TMs and establishment of the active TM6-TM6 interface to initiate signaling.

In addition, Liu *et al.,* developed a FRET-based conformation sensor for CaSR through fusion of SNAP-tag at its N terminus of CaSR subunit to label with fluorophores. Their data showed that the CaSR dimer underwent a large conformational change of LB1-LB1 dimer during activation, which indicated that the B-C helix angle rotated from inactive state to active state as the fluorophores was labeled at N-terminus of LB1 domain.

In our inactive structure, the B-C helix angle is about 117° and VFT domain adopts an open-open conformation (Figure 4A, B). The B-C helix angle has rotated approximately 28° from inactive state (117°) to agonist+PAM bound state (89°) and the VFT domain rearranged from open-open configuration to close-close configuration during CaSR activation, consistent with other class C GPCR activation mechanism reported. Therefore, our data suggest that our inactive structure represents the full inactive state. Comparing our inactive open-open conformation with the recently published inactive closed-closed conformation (CaSR^Icc^) revealed similar CR and 7TM domains, but two totally different VFT module conformations, with their closed-closed conformation presenting some characteristics of the active state. Combined with the observations that the B-C helix of CaSR^Icc^, CaSR^Ioc^ and CaSR^Ioo^ states are the same as that of the active state and that closed conformation of VFT is a feature of the active state, it is likely that the recently reported CaSR^Icc^, CaSR^Ioc^ and CaSR^Ioo^ states are several different intermediate states during activation.

We have reported that the crystal structure of CaSR-ECD in the open-open state has the same B-C helix angle as that of the active state. We designated this as an intermediate state (CaSR-ECD^Ioo^) in the manuscript. It is possible that the full inactive state of CasR is in fact several populations which are difficult to capture. The inhibition with NB-2D11 captures the CaSR in its resting open-open state and inhibits the activity of CaSR, indicating that this conformation is a full inactive state of CaSR.

– The authors propose a Calcium binding site that corresponds to one proposed to be essential for CaSR activation by Liu et al., (PNAS 2020), based on models, mutagenesis, and the use of a conformational FRET-based sensor. This work must be cited. Although such a site is very likely to bind Ca ions, the maps provided by the cryo-EM images are far from being sufficient to guarantee that the density observed corresponds to a Ca ion, rather than to another ion. The authors must clearly indicate this and argue why they think this is probably a Ca ion.

We thank the reviewer for pointing out this negligence. We have discussed our newly found calcium binding site with the one that was proposed by Liu et al*.,* (PNAS 2020) in the text.

Liu et al*.,* modeled two functional calcium binding sites in L-amino acid binding cleft and verified that these two binding sites are important for the activation of CaSR using conformational FRET-based sensor and intracellular calcium release assays. Their calcium binding site 1 is composed of S170, D190, Q193, Y218, and E297. Our newly defined bound Ca^2+^ ion is primarily coordinated with the side chains of D190 and E297, the carbonyl oxygen atoms of P188 backbone, and the hydroxyl groups of S170 and Y489. Among them, P188, D190 and S170 are located at LB1 domain, while E297 and Y489 are presented at LB2 domain (Figure 3C, D). The main binding residues S170 and D190 from LB1 and E297 from LB2 are consistent with findings by Liu et al.

We agree with the reviewer that the maps provided by cryo-EM images are insufficient to ensure that the observed density corresponds to a Calcium. We assume that the density represents the presence of Ca^2+^ based on the following reasons. First, from its hexavalent coordination (the coordinating residues P188, D190, S170 and E297, and Y489), this metal is most likely to be Ca^2+^, although another ion cannot be ruled out. Second, we prepared the CaSR sample in a purification buffer supplemented with 10mM Ca^2+^ and without any other bivalent cation, and then applied it for cryo-EM data collection. Third, the mutation of the coordinating residues significantly reduced the effect of Ca^2+^-stimulated intracellular Ca^2+^ mobilization in cells. Fourth, the mutants of the residues (S147A) to bind L-amino acid also largely impaired the Ca^2+^ effect (Geng et al., 2016), which indicates that there should be a Ca^2+^ ion near the L-amino acid because Ca^2+^ ion activates CaSR through the L-amino acid. In addition, the data from Liu *et al.* suggested that S170, D190 and E297 bound with Ca^2+^.

– The relative movement of the 7TM domains observed in these structures also fully agree with what was proposed by Liu et al., (2020) through a Cys cross-linking approach and functional analysis of the cross-linked dimers. This study must be discussed and cited.

We apologize that we didn’t discuss and cite the observation of relative movement of the 7TM domains proposed by Liu et al., (2020). We have discussed and cited the proposition in our text.

Liu *et al.* used a FRET sensor to investigate the 7TM interface rearrangement during the activation of CaSR by a disulfide cross-linking approach and verified that both resting and active interfaces were cross-linked by blot analysis of the SNAP-CaSR subunits. Their result revealed that a relative rearrangement between the two 7TMs during activation, from TM4-TM5 of each subunit facing each other in the inactive state to TM6-TM6 contact in the active state of the dimer.

Our inactive structure reveals that TM5 and TM6 constitute a TM5-TM6/TM5- TM6 plane-plane interface (Figure 7E). The active structure shows a TM6-TM6 interface, contacting at the apex of TM6 helices. To further validate the role of this interface, we showed that mutant of P823R markedly reduce the Ca^2+^-induced receptor activity. The large-scale transition of intersubunit 7TMDs leads to rearrangement of 7TMDs interface from TM5-TM6-plane/TM5-TM6-plane interface to TM6-TM6 mediated interface during the activation of CaSR.

TM6-TM6 interface within our active structure is consistent with the observation of Liu et al. (2020). This contact is considered to be a hallmark of activation in class C GPCR. Our inactive structure shows the TM5-TM6/TM5-TM6 plane-plane interface, consistent of recently reported structure of CaSR in the inactive state (Ling et al., 2021), while Liu *et al.* showed that the 7TMDs interface formed TM4-TM5 conformation in the inactive state of CaSR. The conformational heterogeneity of the interface of 7TMDs in the inactive state indicates that there is a possible dynamic equilibrium between the TM4-TM5 and TM5-TM6 interfaces. For other class C GPCRs, such as the mGluR5 and GABAB receptors, their 7TMDs rearrange from TM5-TM5 interface in the inactive state to TM6-TM6 interface in the active state.

4. The authors should not refer to the agonist+PAM conformation as the active one, as this is very unlikely since the 7TM domain conformation is almost identical to that observed in the nanobody stabilized inactive state. Please refer to this state as the agonist+PAM bound state.

We agree with the reviewer. Following the reviewer’s suggestion, we have referred agonist+PAM conformation as agonist+PAM bound state in the manuscript.

Reviewer #1 (Recommendations for the authors):1. The conclusion that the Ca^2+^ and TCA site are cooperative is based on structural evidence. Have any experiments in the literature shown cooperativity between TCA and Ca^2+^?

Thank you for reminding us. TNCA is a L-tryptophan derivative. Zhang *et al.* solved the crystal structure of human CaSR-ECD and unexpectedly found that TNCA occupied the orthosteric agonist-binding site at the interdomain cleft. Their experiment demonstrated that TNCA had a high affinity for CaSR and potentiated Ca^2+^ activity (Zhang et al., 2016).

We found that TNCA directly activated CaSR in the presence of 0.5mM of Ca^2+^ ions through intracellular Ca^2+^ flux measurement and that this effect on CaSR was concentration-dependent with EC_50_ of 2.839 mM, in agreement with previous reports that L-Trp directly stimulated intracellular Ca^2+^ mobilization in cells stably expressing CaSR using single-cell intracellular Ca^2+^ microfluorimetry (Conigrave et al., 2004; Conigrave et al., 2000; Geng et al., 2016; Rey et al., 2005; Young and Rozengurt, 2002)

**Author response image 1. sa2fig1:** Does-response curves of TNCA-induced intracellular Ca^2+^ mobilization in presence of 0. 5mM extracellular Ca^2+^ ion.

2. The nanobody is an important part of this story. It would be helpful to provide more information on the selection of the nanobody. Was the inactive conformation the target of the nanobody selection? Was anything done during the screening to select for a nanobody against the inactive conformation? I suggest introducing the nanobody selection and activity assays earlier in the manuscript.

We appreciate the reviewer’s great suggestion. We have added a paragraph (Identification of camelid nanobodies stabilizing the inactive state of CaSR)

at the start of the Results section in our manuscript to describe the selection and activity assays of the nanobody.

For structural studies, we used nanobody to stabilize CaSR in the inactive conformation. Published structures of CaSR-ECD demonstrate that agonist binding induces conformational changes of VFT model of CaSR, whereby two separate LB2 domains approach each other, forming a novel interface in the active state (Geng et al., 2016). Based these structural information, we introduced a potential N-linked glycosylation site on the contacting interface between LB2 domains in the active CaSR to block the interaction of LB2 domains and keep the CaSR in an inactive state. We made a double mutation R227N-E229S at the dimer interface of LB2 domain to introduce N-linked glycosylation at 227 residues site. We immunized two camels with the mutant of CaSR and generated nanobody phage display library. We performed two rounds of bio-panning on the mutant of CaSR, and used enzyme-linked immunosorbent assay (ELISA) to verify the nanobodies that specifically bound to CaSR. We performed intracellular Ca^2+^ flux assay to determine whether screened nanobodies could stabilize CaSR in the inactive state. Out of several CaSR binders, NB-2D11 and NB88 significantly inhibited the activity of CaSR with IC_50_ of 41.7 nM and 167.1 nM, respectively (Figure 1A, B). Using Surface Plasmon Resonance (SPR) to measure binding kinetics, both nanobodies NB-2D11 and NB88 demonstrated high-affinity binding to CaSR with K_D_ of 0.24 nM and 3.9 nM, respectively (Figure 1C, D). We then selected NB-2D11, which has a greater binding affinity of the two nanobodies, for structural study.

3. The authors should elaborate on their rationale for arguing that the current structures are more representative of the inactive state than prior models. Why is the prior model of the inactive state described as "controversial?"

We have addressed this concern in our earlier response. Please refer to Essential Revisions #3.

4. The quality of the model (bonds, angles, C-β deviations, etc) should be evaluated using a web server like MolProbity to ensure that the model statistics are satisfactory. These should be reported.

Thank you for your suggestion. The quality of the model has been evaluated using MolProbity. The data is reported as Table 1. Cryo-EM data collection, refinement and validation statistics**.**

Reviewer #2 (Recommendations for the authors):1 – The authors propose a Calcium binding site that corresponds to one proposed to be essential for CaSR activation by Liu et al. (PNAS 2020), based on models, mutagenesis, and the use of a conformational FRET-based sensor. This work must be cited. Although such a site is very likely to bind Ca ions, the maps provided by the cryo-EM images are far from being sufficient to guarantee that the density observed corresponds to a Ca ion, rather than to another ion. The authors must clearly indicate this and argue why they think this is probably a Ca ion.

We have addressed this concern in our earlier response. Please refer to Essential Revisions #3.

2 – The authors should not refer to the agonist+PAM conformation as the active one, as this is very unlikely since the 7TM domain conformation is almost identical to that observed in the nanobody stabilized inactive state. I recommend they refer to this state as the agonist+PAM bound state.

We agree with the reviewer. Following the reviewers’s suggestion, we have referred agonist+PAM conformation as agonist+PAM bound state in the manuscript.

3 – Most of the inactive structures of the isolated CaSR VFT dimers are all in an orientation corresponding to the active one of the mGlu receptors, with both lobes 2 being in close contact. The recent full length structures revealed three possible inactive states, the VFT dimer in the resting closed closed (Rcc), Resting open closed (Rco) or resting open-open (Roo). Here, the authors report a conformation corresponding to the Roo state that corresponds to the inactive state of mGluR VFTs, but they do not provide a clear demonstration that this is indeed the case. One argument that can be used is based on the work by Liu et al., (2020) that show that the CaSR receptor VFT dimer undergoes a similar change in its orientation during activation, as that of the mGlu receptors. Another argument will be the analysis of the functional effect of the identified nanobody. It is shown to inhibit the receptor activity (Figure 5b), but can a constitutive activity of the CaSR receptor resulting from the other resting conformation be detected? If so, can it be inhibited by NB-2D11?

We have addressed most of this concern in our earlier response. Please refer to Essential Revisions #3.

Thanks for your interesting question regarding the functional effect of our identified nanobody. It is well known that the CaSR interacts with the G_q/11_ and G_i/o_. Our experiment shows that NB-2D11 can inhibit the Ca^2+^_o_-induced Ca^2+^_i_ release through the G_q/11_ mediated activity. We sought to investigate whether NB-2D11 could have the inhibitory effect on the Ca^2+^-induced activity of CaSR through the G_i/o_ pathway using cAMP accumulation assay. CaSR-dependent cAMP inhibition was measured using the cAMP Dynamic 2 Assay Kit purchased from Cisbio Bioassays. Our result showed the Ca^2+^-mediated decrease in cAMP accumulation was abrogated by the incubation with NB-2D11.

**Author response image 2. sa2fig2:** Normalized cAMP accumulation upon stimulation with 10 mM ca^2+^ in the absence or presence of NB2D11.

4 – The relative movement of the 7TM domains observed in these structures also fully agree with what was proposed by Liu et al., (2020) through a Cys cross-linking approach and functional analysis of the cross-linked dimers, with dimers cross-linked through TM6 being constitutively active, while Ca effect was dramatically reduced in TM4 or 5 cross-linked dimers. This study must be discussed and cited.

We have addressed this concern in our earlier response. Please refer to Essential Revisions #3.

5 – The authors must also be clear on the relative role of TNCA (or other L-AA) and Ca ions on the activation of the receptor. Using a well-controlled assay, Liu et al. (2020) reported that Ca ions are sufficient to activate the CaSR, even in the absence of L-AA, such that Ca ion can be considered as the agonist of the CaSR. In contrast, any tested L-AA did not activate the receptor, but largely potentiated the Ca effect, such that these L-AA should be considered as PAMs of the CaSR. This must be clarified in the text.

The reviewer’s question is very important. It is also a problem that has puzzled me and made me think for several years.

Liu et al*.,* (2020) reported that Ca^2+^ are sufficient to activate the CaSR using a well-controlled assay, even in the absence of L-AA, such that Ca^2+^ can be considered as the agonist of the CaSR.

At present, the prevailing view is that the principal agonist of CaSR is extracellular Ca^2+^ (Hofer and Brown, 2003). The L-amino acids, such as L-Trp, can enhance the sensitivity of CaSR toward Ca^2+^ ions (Conigrave et al., 2000), and are considered as positive allosteric modulators of the receptor (Saidak et al., 2009). However, as L-amino acids or their analogs are endogenous agonists of other class C GPCRs, this is somewhat inconsistent from the perspective of GPCR classification and evolution. Previous studies have also shown that if Ca^2+^ concentration is higher than the threshold of 0.5mM, L-amino acid can activate the receptor (Conigrave et al., 2004; Conigrave et al., 2000; Rey et al., 2005; Young and Rozengurt, 2002). Therefore, it suggested that Ca^2+^ and L-amino acid can act as co-agonists of the receptor (Conigrave et al., 2000; Young and Rozengurt, 2002). Using single-cell intracellular Ca^2+^ microfluorimetry, L-Trp has been shown to directly stimulate intracellular Ca^2+^ mobilization in cells stably expressing CaSR, with its efficacy and potency increase with increases in concentration of Ca^2+^ ions, hence providing direct evidence that L-amino acids are agonists of CaSR (Geng et al., 2016). However, this view has yet to be widely accepted because it is difficult to observe L-amino acids directly activating CaSR. The mode of action of Ca^2+^ ion and L-amino acids on CaSR remains controversial.

A few years ago, when we prepared human CaSR samples without L-amino acid for crystallization, we obtained crystal structure of CaSR ECD with a stretch of continuous density in the interdomain cleft. Unfortunately, we were not able to identify the structure of this ligand (Geng et al., 2016). Zhang et al., (2016) solved the crystal structure of human CaSR-ECD in absence of L-AA, and found density at the interdomain cleft, too. It was identified as TNCA. Moreover, they found that TNCA had a high affinity for CaSR and potentiated Ca^2+^ activity (Zhang et al., 2016). Ling et al., (2021) found a clear extra density in both VFT domain of both structures of CaSR^Icc^ and CaSR^Ca^ in the absence of L-amino acid. It indicated that ambient L-amino acids (or L-amino acid derivative, such as TNCA) had bound to CaSR as an endogenous ligand (Ling et al., 2021). Our agonist +PAM bound structure shows that TNCA and a newly identified Ca^2+^ ion bind at the interdomain cleft of the VFT module, and they both interact with both LB1 and LB2 domains to facilitate extracellular domain closure (Figure 3A), therefore forming the closed state of the ligand binding domain required for CaSR activation. This indicates that both TNCA and the Ca^2+^ ion contribute to the activation of CaSR. TNCA and the bound Ca^2+^ share three common binding residues S170 and D190 of LB1 domain and E297 from LB2 domain (Figure 3E, F), which suggest that the CaSR is synergistically activated by TNCA and Ca^2+^ ions.

Our experiments have shown that TNCA directly stimulate intracellular Ca^2+^ mobilization in cells stably expressing CaSR (Figure 3G), suggesting that TNCA are agonists of CaSR. The structure of CaSR shows that TNCA binds at the cleft between LB1 and LB2 domains, which is the conserved ligand binding site for all class C GPCR agonists (Geng et al., 2013; Geng et al., 2016; Kunishima et al., 2000; Muto et al., 2007; Tsuchiya et al., 2002), and TNCA has a binding pattern similar to that of the endogenous agonists of mGluR and GABAB receptors (Figure 3D) (Geng et al., 2016). The key coordination residues are very conserved, such as S147 and S170 (Geng et al., 2013; Geng et al., 2016; Kunishima et al., 2000; Zhang et al., 2016). Moreover, TNCA (or L-Trp) interact with residues from LB1 and LB2 to stabilize the closure of VFT, and the signal mutation of the contacting residues (T145I, S147A, S170A, Y218S, E297K) substantially reduce the function of the receptor, even if some of these residues (S147A, T145I and Y218S) are not related to the coordinating residues of Ca^2+^, hence indicating that TNCA or L-Trp plays a key role in the activation of CaSR (Geng et al., 2016). In addition, Ling *et al.* (2021) tried to determine the cryo-EM structures of CaSR in the presence of a high concentration of Ca^2+^ to address the question of whether Ca^2+^ ions alone can activate CaSR in the absence of L-Trp. Three different 3D models were obtained, in which the VFT adopted closed-closed, closed-open and open-open conformations. They observed densities located at the interdomain crevice of the VFT module in the closed state. They suggested that L-amino acids or its derivate is bound to CaSR, because the densities were much larger than that of a Ca^2+^ ion. However, they did not obtain the closed conformation of VFT containing only Ca^2+^ ion between the cleft (Ling et al., 2021). The results indicate that Ca^2+^ ion alone is not enough to induce the closure of the VFT module even in the presence of a high concentration of Ca^2+^ ion, and L-amino acid or its derivate is required to stabilize the closed conformation of VFT module. Altogether, L-amino acids are the endogenous agonists of CaSR, in agreement with that of other class C GPCRs.

Three different groups prepared CaSR samples without L-amino acids or derivatives for crystal or cryo-EM structural study, but they all unexpectedly obtained the active structure of CaSR or CaSR ECD with the closed-closed conformation of VFT module containing undefined ligands (Geng et al., 2016; Ling et al., 2021; Zhang et al., 2016). The ligand was identified as TNCA, which had a high affinity for CaSR and potentiated Ca^2+^ activity (Zhang et al., 2016). As we all know, it takes a long time to purify CaSR in TNCA-free buffer for structural study, nevertheless, the endogenous TNCA would still bind to CaSR. This indicates that TNCA has a high affinity for CaSR or that it is buried in closed VFT module such that it is difficult to be washed off.

Gentle washing is impossible to remove TNCA bound to CaSR in various function assays. In the absence of TNCA (or L-amino acid), it is possible that TNCA would still bind to CaSR to stabilize the closed conformation of VFT, although the control assay is well done. Ca^2+^ ions seem to directly activate the CaSR alone.

We speculate that as the endogenous TNCA exists and is difficult to be replaced by other added L-amino acids, it is challenging to observe the direct activation of other L-amino acids, or the observed allosteric regulation is also a comprehensive result when we performed function assay. These highlight the need for further experiments investigating the kinetics and dynamics of L-amino acid or TNCA binding to CaSR to strengthen our findings, however, such the magnitude of the experiments involved are beyond the scope of the current paper.

In addition to the role of Ca^2+^ ion binding at the cleft of VFT module discussed above to stabilize the closed conformation of VFT, another role of the Ca^2+^ ions should be considered, in which Ca^2+^ ion is coordinated by D234, E231 and G557, and bridges the LB2 domain of one subunit and CR domain of the second domain. The Ca^2+^ ion facilitates the formation of the active conformation of CaSR, which is the reason that the L-amino acids (or TNCA) activate CaSR in the presence of Ca^2+^ ion above the threshold concentration of 0.5mM.

6 – The intermediate state presented should be better explain in the text.

Thank you for your suggestion. We have discussed and explained in detail the intermediate state in the Results section. Based on the rotation of the angles of B-C Helix during the activation of CaSR as discussed above (response to Essential Revisions #3), the orientation of the LB1-LB1 dimer interface of the open-open conformation of CaSR_ECD_ crystal structure is similar to that of agonist+PAM bound state, which is different from that of our full inactive state of CaSR, indicating that the open-open conformation of CaSR_ECD_ crystal structure presents some characteristics of the conformation of the active state. In addition, the superimposition of different state of VFT domain shows that the distances between C-termini of VFT domain of CaSR gradually decreases from the full active state to active state. The distance between C-termini of VFT domains of the open-open conformation of CaSR_ECD_ crystal structure is between full inactive and active states. Therefore, we defined the open-open conformation of CaSR_ECD_ crystal structure as one of the intermediate states that are ready for binding of L-AA and calcium ions.

7 – Please be more specific in Figure 4 on how the distances between the C-termini of the VFTs were measured. Indicate which atom was used as a reference.

Thank you for your question. We have measured the distances of Cα atom of N541 at the C-termini of the VFTs.

Reviewer #3 (Recommendations for the authors):As discussed above, I believe the manuscript could be strengthened by better discussing the distinct aspects of the structures compared to the Ling et al., published paper.

We have addressed this concern in our earlier response. Please refer to Essential Revisions #3. In our study, we have determined the cryo-EM structures of CaSR in the inactive and the agonist+PAM bound states. The overall conformation of our agonist+PAM bound structure is almost identical to that of the structure of closed-closed conformation of Ca^2+^/Trp bound state (CaSR^Acc^ and CaSR^Ca^). We compare our inactive open-open conformation with the recently published inactive closed-closed conformation, which revealed a similar CR domain and 7TM domain that is separated in these inactive states, but the VFT module reveal the totally different conformation, the closed-closed conformation of their structure presents some characteristics of the active state. It indicates that the CaSR in the inactive state has conformational heterogeneity. In other words, it suggests that in addition to the full inactive state and the active state, there are multiple intermediate states in the process of activation.

Also I would recommend the authors to go through the manuscript again and to double check typos and grammatical errors. I have noted a few here:L9: our data showsL33: through which can be removed.L44: the active structure has demonstrated a calcium…I would rather say the active structure presents a calcium…This is actually the case several times in the manuscript (L57, 65…). When speaking about structural or map features, I would suggest to replace "has demonstrated" or "have demonstrated" or 'has shown' by "present" or similar.L78: the sentence is not clear and should be rewritten:and the closed conformation of the VFT module has shown in the active state, which is stabilized by the TNCA binding at the VFT module and 3 distinct calcium binding sites within each ECDL102: throughout the mansucript you wrote promoter instead of protomer.L231: is consisted (consistent)L240: contacting with the map does not make sense in the sentence.L254: presents similar, except …similar what? This sentence should be written in a clearer way.L290: There is a common interface constitute of residues 759-763 at ECL2 and residues 601-604 at the C-terminal of CRD (Figure 7A), and the interface in the active conformation is more compact than that of in the inactive conformationThis sentence should be written in a clearer way.L301: relay should be relays. I would however write 'ECL2 seems to play a key role in relaying…..'L322-325: This sentence should be rewritten, as it is not clear and it contains grammatical errors (allows).L343-344: thirdly, the closure of VFT that stabilized by the agonist and Ca^2+^ ion is a landmark event during the CaSR activation.You should add 'is' before satbilized.L348-349: This sentence should be written in a clearer way.L374: same error as before with consisted instead of consistent.

We appreciate the referee for pointing out and correcting grammatical mistakes and typos. The above errors have been corrected in the manuscript.

References

Conigrave, A.D., Mun, H.C., Delbridge, L., Quinn, S.J., Wilkinson, M., and Brown, E.M. (2004). L-amino acids regulate parathyroid hormone secretion. *J Biol Chem* 279, *38151-38159.*https://doi.org/10.1074/jbc.M406373200

Conigrave, A.D., Quinn, S.J., and Brown, E.M. (2000). L-amino acid sensing by the extracellular ca^2+^-sensing receptor. *Proceedings of the National Academy of Sciences of the United States of America* 97, *4814-4819.*https://doi.org/10.1073/pnas.97.9.4814

Geng, Y., Bush, M., Mosyak, L., Wang, F., and Fan, Q.R. (2013). Structural mechanism of ligand activation in human GABA(B) receptor. *Nature* 504, *254-259.*https://doi.org/10.1038/nature12725

Geng, Y., Mosyak, L., Kurinov, I., Zuo, H., Sturchler, E., Cheng, T.C., Subramanyam, P., Brown, A.P., Brennan, S.C., Mun, H.C.*, et al.* (2016). Structural mechanism of ligand activation in human calcium-sensing receptor. *eLife* 5https://doi.org/10.7554/*eLife*.13662

Hofer, A.M., and Brown, E.M. (2003). Extracellular calcium sensing and signalling. *Nat Rev Mol Cell Biol* 4, *530-538.*https://doi.org/10.1038/nrm1154

Kunishima, N., Shimada, Y., Tsuji, Y., Sato, T., Yamamoto, M., Kumasaka, T., Nakanishi, S., Jingami, H., and Morikawa, K. (2000). Structural basis of glutamate recognition by a dimeric metabotropic glutamate receptor. *Nature* 407, *971-977.*https://doi.org/10.1038/35039564

Ling, S., Shi, P., Liu, S., Meng, X., Zhou, Y., Sun, W., Chang, S., Zhang, X., Zhang, L., Shi, C.*, et al.* (2021). Structural mechanism of cooperative activation of the human calcium-sensing receptor by Ca(2+) ions and L-tryptophan. *Cell research*https://doi.org/10.1038/s41422-021-00474-0

Mos, I., Jacobsen, S.E., Foster, S.R., and Brauner-Osborne, H. (2019). Calcium-Sensing Receptor Internalization Is β-Arrestin-Dependent and Modulated by Allosteric Ligands. *Molecular pharmacology* 96, *463-474.*https://doi.org/10.1124/mol.119.116772

Muto, T., Tsuchiya, D., Morikawa, K., and Jingami, H. (2007). Structures of the extracellular regions of the group II/III metabotropic glutamate receptors. *Proceedings of the National Academy of Sciences of the United States of America* 104, *3759-3764.*https://doi.org/10.1073/pnas.0611577104

Rey, O., Young, S.H., Yuan, J., Slice, L., and Rozengurt, E. (2005). Amino acid-stimulated ca^2+^ oscillations produced by the ca^2+^-sensing receptor are mediated by a phospholipase C/inositol 1,4,5-trisphosphate-independent pathway that requires G12, Rho, filamin-A, and the actin cytoskeleton. *J Biol Chem* 280, *22875-22882.*https://doi.org/10.1074/jbc.M503455200

Saidak, Z., Boudot, C., Abdoune, R., Petit, L., Brazier, M., Mentaverri, R., and Kamel, S. (2009). Extracellular calcium promotes the migration of breast cancer cells through the activation of the calcium sensing receptor. *Exp Cell Res* 315, *2072-2080.*https://doi.org/10.1016/j.yexcr.2009.03.003

Tsuchiya, D., Kunishima, N., Kamiya, N., Jingami, H., and Morikawa, K. (2002). Structural views of the ligand-binding cores of a metabotropic glutamate receptor complexed with an antagonist and both glutamate and Gd3+. *Proceedings of the National Academy of Sciences of the United States of America* 99, *2660-2665.*https://doi.org/10.1073/pnas.052708599

Young, S.H., and Rozengurt, E. (2002). Amino acids and ca^2+^ stimulate different patterns of ca^2+^ oscillations through the ca^2+^-sensing receptor. *Am J Physiol Cell Physiol* 282, *C1414-1422.*https://doi.org/10.1152/ajpcell.00432.2001

Zhang, C., Zhang, T., Zou, J., Miller, C.L., Gorkhali, R., Yang, J.Y., Schilmiller, A., Wang, S., Huang, K., Brown, E.M.*, et al.,* (2016). Structural basis for regulation of human calcium-sensing receptor by magnesium ions and an unexpected tryptophan derivative co-agonist. *Science advances* 2, *e1600241.*https://doi.org/10.1126/sciadv.1600241